# What matters for Representation Alignment: Global Information or Spatial Structure?

**Jaskirat Singh**[1][2]  **Xingjian Leng**[2]  **Zongze Wu**[1]
**Liang Zheng**[2]  **Richard Zhang**[1]  **Eli Shechtman**[1]  **Saining Xie**[3]
[1]Adobe Research  [2]ANU  [3]New York University

## Abstract

Representation alignment (REPA) guides generative training by distilling representations from a strong, pretrained vision encoder to intermediate diffusion features. We investigate a fundamental question: what aspect of the target representation matters for generation, its *global* semantic information (e.g., measured by ImageNet-1K accuracy) or its spatial structure (i.e. pairwise cosine similarity between patch tokens)? Prevalent wisdom holds that stronger global semantic performance leads to better generation as a target representation. To study this, we first perform a large-scale empirical analysis across 27 different vision encoders and different model scales. The results are surprising — spatial structure, rather than global performance, drives the generation performance of a target representation. To further study this, we introduce two straightforward modifications, which specifically accentuate the transfer of *spatial* information. We replace the standard MLP projection layer in REPA with a simple convolution layer and introduce a spatial normalization layer for the external representation. Surprisingly, our simple method (implemented in <4 lines of code), termed iREPA, consistently improves convergence speed of REPA, across a diverse set of vision encoders, model sizes, and training variants (such as REPA, REPA-E, Meanflow, JiT etc). Our work motivates revisiting the fundamental working mechanism of representational alignment and how it can be leveraged for improved training of generative models.

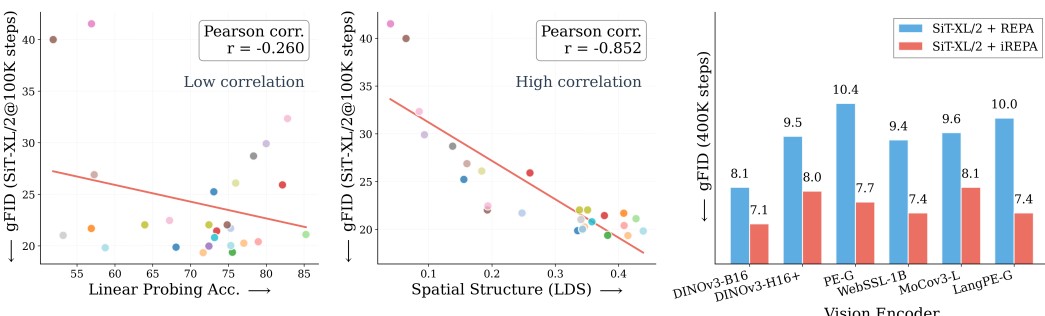

Figure 1: **What matters for representation alignment?** *Left:* Correlation analysis across 27 diverse vision encoders. Surprisingly, contrary to the prevailing wisdom, we find that spatial structure, rather than global performance (measured by linear probing accuracy), drives the generation performance of a target representation. *Right:* We further study this by introducing two simple modifications to accentuate the transfer of spatial features from target representation to diffusion model. Our simple approach consistently improves the convergence speed of REPA across diverse settings.

## 1 Introduction

Representation alignment has emerged as a powerful technique for accelerating the training of diffusion transformers ([Ma et al., 2024](#); [Peebles & Xie, 2023](#)). By aligning internal diffusion

---

*Done during internship at Adobe Research

**Project:** https://end2end-diffusion.github.io/irepa

Figure 2: **Motivating examples — spatial structure matters.** Metrics comparison showing inverse relationship between ImageNet accuracy and generation quality. *Left:* PE-G, despite having significantly higher validation accuracy (82.8% vs. 53.1%), shows worse generation quality compared to SpatialPE-B (Bolya et al., 2025). *Right:* Similarly, WebSSL-1B (Fan et al., 2025) also shows much better global performance (76.0% vs. 53.1%), but worse generation. **Spatial Self-Similarity:** We find that spatial structure instead provides a better predictor of generation quality than global performance. See §3 for spatial structure metric. All results reported at 100K using SiT-XL/2 and REPA.

representations with pretrained self-supervised visual encoders, recent methods have demonstrated significant improvements in the convergence speed and final performance (Yu et al., 2024; Leng et al., 2025a). However, despite these empirical successes, there remains limited understanding of the precise mechanisms through which self-supervised features enhance diffusion model training. A fundamental question persists: is the improvement primarily driven by incorporating better global semantic information, as commonly measured through linear probing performance, or does it stem from better capturing spatial structure, characterized by the relationships between patch token representations?

Understanding these mechanisms is crucial for advancing generative model training, as it directly impacts one's ability to select the optimal target representation and maximize its benefits. Currently, a prevalent assumption is that encoder performance for representation alignment correlates strongly with ImageNet-1K validation accuracy, a proxy measure of global semantic understanding (Oquab et al., 2024; Chen et al., 2021). That is, target representations with better ImageNet performance are hypothesized to lead to better generation (Yu et al., 2024). As such, increases in linear probing accuracy of diffusion features are frequently cited as evidence for the effectiveness of representation alignment, reinforcing the emphasis on global semantic information as the primary driver of improvement. The following quote from REPA (Yu et al., 2024) captures the current understanding:

> *"When a diffusion transformer is aligned with a pretrained encoder that offers more semantically meaningful representations (i.e., better linear probing results), the model not only captures better semantics but also exhibits enhanced generation performance, as reflected by improved validation accuracy with linear probing and lower FID scores."*

In this paper, we challenge this conventional wisdom by systematically investigating what truly drives representation alignment: global semantic information or spatial structure? Through large-scale empirical analysis across diverse vision encoders, including recent large vision foundation models such as WebSSL (Fan et al., 2025), DINOv3 (Siméoni et al., 2025), perceptual encoders (Bolya et al., 2025), C-RADIO (Heinrich et al., 2024), we uncover 3 surprising findings.

**Higher validation accuracy does not imply better representation for generation.** Contrary to prevailing assumptions, vision encoders with higher global semantic performance, measured by ImageNet-1K linear probing accuracy, can often underperform for generation. For instance, consider PE-Spatial-B, a spatially-tuned model derived from PE-Core-G (Bolya et al., 2025). We find that while PE-Spatial-B shows a much worse validation accuracy[1] on patch tokens than PE-Core-G (53.1% vs. 82.8%), it leads to better generation with REPA (Figure 2). Similarly, WebSSL-1B (Fan et al., 2025) shows much better global performance (76.0% vs. 53.1%), but worse generation. In Section §2, we find that this pattern holds across various target representations, suggesting a fundamental principle in how representation alignment benefits diffusion training.

**Spatial structure not global information determines generation performance.** To quantify this insight, we consider several straightforward metrics to measure the spatial self-similarity structure

---

[1]Similar to REPA (Yu et al., 2024), we use linear probing accuracy on patch tokens to measure global semantic performance of external representation as only patch tokens are used for representation alignment.

(Shechtman & Irani, 2007) between patch tokens (§3). We then perform large-scale analysis computing the correlation between generation FID with REPA, linear probing accuracy, and spatial structure across 27 vision encoders and 3 model sizes (SiT-B, SiT-L, SiT-XL). Surprisingly, all spatial structure metrics exhibit remarkably better correlation (Pearson $|r| > 0.852$) with generation FID scores, far exceeding the predictive power of ImageNet-1K validation accuracy ($|r| = 0.26$). These findings are also supported by additional experiments showing that external representations with very limited global information can be used to get significant gains with REPA. For instance, SAM2 leads to better generation with REPA than other encoders with $\sim 60\%$ higher ImageNet accuracy (Fig. 3). Similarly, while not the best, we find that classical spatial features such as SIFT (Lowe, 1999) and HOG (Dalal & Triggs, 2005) can also be used to achieve decent gains with representation alignment (§3).

**Accentuating spatial features improves convergence performance.** To further study this, we next introduce two straightforward modifications ($<4$ lines of code), which specifically accentuate the transfer of "spatial" information from target representation to diffusion model: (1) We first introduce a spatial regularization layer which boosts the spatial contrast of the target representations. (2) Next, we show that the standard MLP-based projection layer (used to map diffusion features to target representation dimensions) causes loss of spatial information (Figure 6a). To avoid this, we replace it with a simple convolution layer. The results are surprising: our simple method, termed iREPA, consistently improves convergence speed of REPA across diverse variation in encoders, model sizes, and training recipes *e.g.*, REPA-E (Leng et al., 2025a) and Meanflow (Geng et al., 2025) w/ REPA.

We highlight the main contributions of this paper below:

- We perform large-scale empirical analysis showing that spatial structure and not global semantic information drives the effectiveness of representation alignment.
- We introduce a straightforward and fast-to-compute Spatial Structure Metric (SSM), which shows significantly higher correlation with downstream FID performance than linear probing scores.
- We propose simple modifications to better accentuate the transfer of spatial information from the target representation to diffusion features. Our simple method shows consistent improvements in the convergence speed over REPA across variations in target representation, model architectures as well as training recipe variants (REPA, REPA-E, Meanflow w/ REPA, JiT w/ REPA etc).

## 2  MOTIVATION: GLOBAL INFORMATION MATTERS LESS

We first provide several motivating examples showing that a target representation with higher global performance (ImageNet-1K accuracy) does not imply better generation performance with REPA. We later show in §3, that these previously-unexplained observations can instead be better explained by measuring the spatial structure of the target representation.

**Recent vision encoders.** We examine different recent vision encoders, including Perceptual Encoders (Bolya et al., 2025), WebSSL (Fan et al., 2025), and DINOv3 (Siméoni et al., 2025). Consider PE-Spatial-B (80M), a small spatially tuned model derived from PE-Core-G (1.88B) (Bolya et al., 2025). As seen in Figure 2, we find that while PE-Core-G achieves much higher ImageNet-1K accuracy (82.8% vs 53.1%), it performs worse when used as the target representation for REPA (FID 32.3 vs 21.0). Similarly, WebSSL-1B (1.2B) achieves much higher ImageNet-1K accuracy (76.0% vs 53.1%) but performs worse when used as target representation for REPA (FID 26.1 vs 21.0).

**SAM2 outperforms vision encoders with much higher ImageNet-1K accuracy.** To further understand how little global information impacts generation performance, we analyze SAM2-S vision encoder (46M) (Ravi et al., 2024) — a small model with very little global information and validation accuracy of only 24.1% (Fig. 3a). Surprisingly, when used for REPA, SAM2-S achieves better FID than other vision encoders with significantly higher ImageNet-1K accuracy *e.g.*, PE-Core-G (82.8%).

**Larger models within same encoder family can have similar or worse generation performance.** A common perception is that larger models within the same encoder family have better representations (measured by ImageNet-1K accuracy). However, for representation alignment, larger model variants often lead to similar (DINOv2) or even *worse* (PE, Cradio) generation performance (Fig. 3b). Notably (Yu et al., 2024) also make a similar observation for DINOv2 and explain it as "we hypothesize is due to all DINOv2 models being distilled from the DINOv2-g model and thus sharing similar representations". We later show that this trend can be better explained using spatial structure (§3).

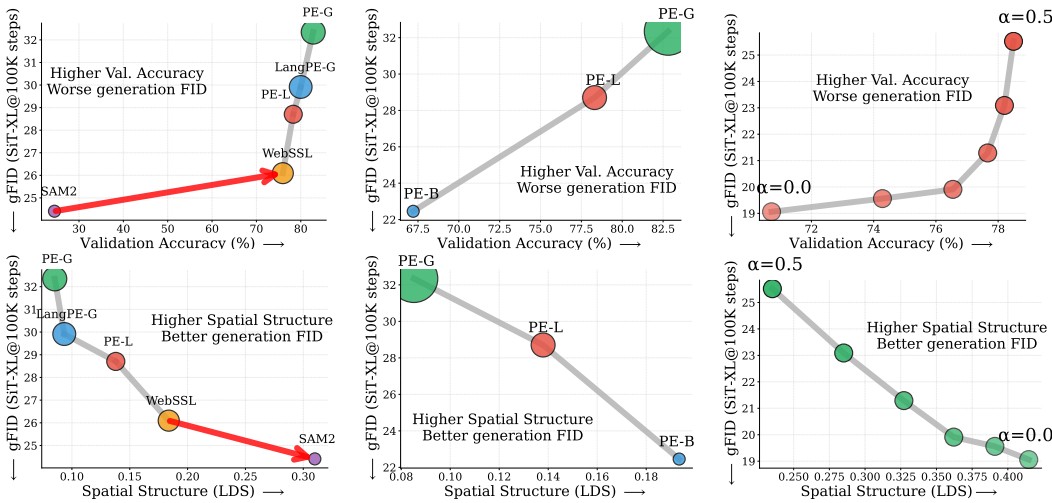

(a) SAM2 vs "better" encoders.  (b) Larger models but worse gFID.  (c) Adding global info. hurts FID.

Figure 3: **Higher global information does not imply better REPA performance. Top row:** Several trends showing global performance does not correlate well with generation FID when using REPA. (a) SAM2-S with only validation accuracy of 24.7% results in better generation performance with REPA compared with other models with $\sim 60\%$ higher validation accuracy. (b) Larger encoders within same family can have better validation but worse generation performance. (c) Adding global information to patch tokens via CLS token improves global performance but hurts generation. **Bottom row:** We show that spatial structure rather then global performance provides a better indicator for generation. Please see §3 for large-scale detailed analysis across different spatial structure metrics. All results are reported w/o classifier free guidance, SiTXL/2 w/ REPA and 250 NFE (Yu et al., 2024) for inference.

**Adding more global information can hurt generation.** To test whether additional global information benefits representation alignment, we conduct controlled experiments that inject global semantics, using the CLS token, into local patch tokens (with DINOv2 as encoder). The CLS token mixing operation is $\mathbf{p}_i^{\text{new}} = \mathbf{p}_i + \alpha \cdot \mathbf{c}$, where $\mathbf{p}_i$ denotes the $i^{\text{th}}$ patch token, $\mathbf{c}$ the CLS token, and $\alpha \in [0, 0.5]$ controls the mixing strength. As shown in Figure 3c, as $\alpha$ increases from 0 to 0.5, linear probing accuracy improves monotonically from 70.7% to 78.5%. However, generation quality deteriorates significantly, with FID scores worsening from 19.2 at $\alpha = 0$ to 25.4 at $\alpha = 0.5$.

The above observations highlight that global performance of a representation is not a good indicator of its REPA performance. We next show (§3) spatial structure instead provides a better signal.

> • **Finding 1.** *Better global semantic information (e.g., ImageNet-1K accuracy) does not imply a better representation for generation.*

## 3  SPATIAL STRUCTURE MATTERS MORE

We hypothesize that spatial structure, rather than global information, drives the generation performance with a target representation. To quantify this, we first consider several straightforward metrics to measure the spatial self-similarity structure (Shechtman & Irani, 2007) of target representations. We then show that all spatial structure metrics not only correlate much higher with gFID than ImageNet-1K accuracy, but can be also used to explain previously unexplained observations in §2.

**Measuring spatial self-similarity structure.** Given an image $\mathcal{I}$ and vision encoder $\mathcal{E}$, we define:

• *Self-similarity*. Let $\mathcal{X} = \mathcal{E}(\mathcal{I}) \in \mathbb{R}^{T \times D}$ be the extract patch representations, with $T = H \times W$ patches. Let kernel $K_\mathcal{X}(\cdot, \cdot) \in \mathbb{R}$ measure appearance self-similarity (Shechtman & Irani, 2007) between patch tokens. Here, we use the cosine kernel $K_\mathcal{X}(t, t') = \frac{\langle \mathbf{x}_t, \mathbf{x}_{t'} \rangle}{\|\mathbf{x}_t\|_2 \|\mathbf{x}_{t'}\|_2}$.

• *Spatial distance*. Let $\mathcal{P} \in \mathbb{N}^{T \times 2}$ be the spatial location of each of the $T$ tokens in coordinate space and $d(\cdot, \cdot) \in \mathbb{N}$ be the Manhattan distance between pairs of tokens.

We then measure *spatial* self-similarity metric as how self-similarity $K_\mathcal{X}$ varies with lattice distance $d$ between patch tokens. Intuitively, larger values indicate stronger spatial organization (closer patches

Figure 4: **Spatial structure shows higher correlation with generation quality than linear probing.** Correlation analysis across 27 diverse vision encoders, SiT-XL/2 and REPA. Linear probing shows weak correlation with FID (Pearson $|r| = 0.260$), while all spatial structure metrics: LDS ($|r| = 0.852$), SRSS ($|r| = 0.885$), CDS ($|r| = 0.847$), and RMSC ($|r| = 0.888$), demonstrate much stronger correlation with generation performance. See Fig. 10 for detailed plots with encoder labels.

more similar to each other than patches further away). By default, we use a simple correlogram contrast (local *vs.* distant) metric (Huang et al., 1997):

$$\text{LDS}(\mathcal{X}, \mathcal{P}) = \mathbb{E}_{d(t,t') \in (0, r_{\text{near}})} K_{\mathcal{X}}(t, t') - \mathbb{E}_{d(t,t') \geq r_{\text{far}}} K_{\mathcal{X}}(t, t').$$

The final spatial self-similarity score (LDS) is computed as the expectation over patch representation $\mathcal{X}$. We use $r_{\text{near}} = r_{\text{far}} = H/2$ here, though we found correlation to be robust to their exact choices. Please see Appendix B for exact details and alternative spatial metrics (CDS, SRSS, RMSC) which also perform effectively (see Fig. 4).

**Spatial structure correlates much higher with generation performance than linear probing.** We next perform large-scale correlation analysis across 27 diverse vision encoders. As shown in Figure 4, we find that while typically used linear probing shows very weak correlation (Pearson $|r| = 0.26$), all SSM metrics show much higher correlation with generation performance (Pearson $|r| > 0.85$).

**Generalization across model scales.** We verify the correlation across different model scales (SiT-B, SiT-L, SiT-XL) in Figure 5. Linear probing shows weak correlation across model scales ($|r| < 0.306$), while spatial structure shows much higher correlation with generation performance ($|r| > 0.826$).

> • **Finding 2.** *Spatial structure correlates much higher with generation performance than linear probing. We next use spatial metrics to explain previously unexplained trends.*

**Spatial structure can explain previously unexplained trends.** As discussed in §2, global performance (ImageNet-1K accuracy) does not serve as a predictive measure of effectiveness for representation alignment. We find that instead the above spatial metrics serve as better predictors.

(1) *PE-Spatial-B vs PE-Core-G*: Figure 2 shows that PE-Core-G achieves much higher ImageNet-1K accuracy (82.8% vs 53.1%), but performs worse when used as target representation for REPA (FID 32.3 vs 22.0). Looking at spatial structure metric makes things clearer. As seen in Figure 2, despite lower global performance, PE-Spatial-B shows much better spatial structure than PE-Core-G; leading to better generation performance as observed.

(2) *SAM2 outperforms "better" vision encoders*: §2 shows that while SAM2 achieves much lower ImageNet-1K accuracy (24.1%), it leads to better generation than encoders with $\sim 60\%$ higher accuracy. As in Fig. 3a; these gains can be directly explained through SAM2's better spatial structure.

(3) *Larger models in same encoder family underperform*: As shown in Fig. 3b, while larger models in same encoder family show better ImageNet-1K accuracy, they can have worse spatial structure, leading to worse generation performance with REPA. This also aligns with observations from Yu et al. (2024), where they find that larger models can have similar or worse generation performance.

4) *Adding global information to patch tokens via CLS token hurts generation*: In Figure 3c, we observe that increasing global information by mixing the CLS token with patch tokens improves global performance. However, mixing CLS token reduces spatial contrast among patch tokens. This leads to patch tokens having high similarity with otherwise unrelated tokens (*e.g.*, from foreground object to background). The reduced spatial structure thus leads to worse generation performance.

**If spatial structure matters more, can we use SIFT or HOG features for REPA?** Surprisingly, yes. We find that while certainly not the best, classical spatial features like SIFT (Lowe, 1999), HOG (Dalal & Triggs, 2005) and intermediate VGG features (Simonyan & Zisserman, 2014)

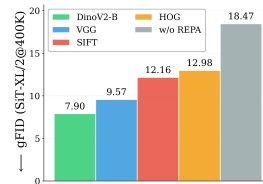

Figure 5: **Correlation analysis across model scales.** Across different model scales, we find that spatial structure (right) consistently shows higher correlation with gFID than linear probing (left).

all lead to performance gains with REPA. This provides further evidence that representation alignment can benefit from spatial features alone without need for additional global information.

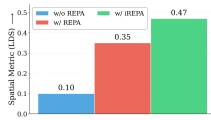

**Can we use spatial metrics to explain gains with REPA?** Yes. As shown, the introduced spatial metrics can be used to explain both gains with REPA as well as our improved training recipe (iREPA) which we introduce next in §4.

# 4 *i*REPA: IMPROVING REPRESENTATION ALIGNMENT BY ACCENTUATING WHAT MATTERS

To further investigate the role of spatial structure in representation alignment, we introduce two straightforward modifications to the original REPA training recipe, which enhance the transfer of spatial features from the teacher (vision encoder) to the student (diffusion transformer) model.

**Convolutional projection layer instead of MLP.** The standard REPA approach uses a 3-layer MLP projection to map diffusion feature dimensions to that of the external representation. However, as shown in Figure 6a, we observe that this projection is lossy and diminishes the spatial contrast between patch tokens. We therefore replace the MLP with a lightweight convolutional layer (kernel size 3, padding 1), that operates directly on the spatial grid. The convolutional structure naturally preserves local spatial relationships through its inductive bias.

**Spatial normalization layer.** Similar to Siméoni et al. (2025), we find that patch tokens of pretrained vision encoders consist of a significant global component. This is evidenced by the high linear probing scores for the mean of patch tokens (Figure 14). Also, we see in Figure 6b that while mixing of this global information (mean of patch tokens) with the patch tokens helps improve global performance, it reduces the spatial contrast between individual patch tokens. This leads tokens (e.g., foreground object) showing high similarity with otherwise unrelated tokens (e.g., background).

Given results from §3, we hypothesize that we can sacrifice this global information (mean of patch tokens) to improve the spatial contrast between the patch tokens. The improved contrast should provide better spatial signal (pairwise similarity between patch tokens) — leading to better REPA performance. To this end, we add a simple spatial normalization layer (Ulyanov et al., 2016) to the patch tokens of the target representation:

$$\mathbf{y} = \frac{\mathbf{x} - \gamma \mathbb{E}[\mathbf{x}]}{\sqrt{\text{Var}[\mathbf{x}] + \epsilon}},$$

where $\mathbf{x} \in \mathbb{R}^{B \times T \times D}$ represents the patch tokens, the expectation and variance are computed across the spatial dimension, and $\epsilon = 10^{-6}$ for numerical stability.

---

**Algorithm 1** Summary of key iREPA modifications to standard REPA training

```
# 1. Conv projection instead of MLP
proj_layer = nn.Conv2d(D_in, D_out, kernel_size=3, padding=1)

# 2. Spatial normalization on encoder features [B, T, D]
x = x - gamma * x.mean(dim=1, keepdim=True)
x = x / (x.std(dim=1, keepdim=True) + 1e- 6)
```

---

## 4.1 EXPERIMENTS

In this section, we validate the performance of the improved training recipe through extensive experiments on Imagenet $256 \times 256$. In particular, we investigate the following research questions:

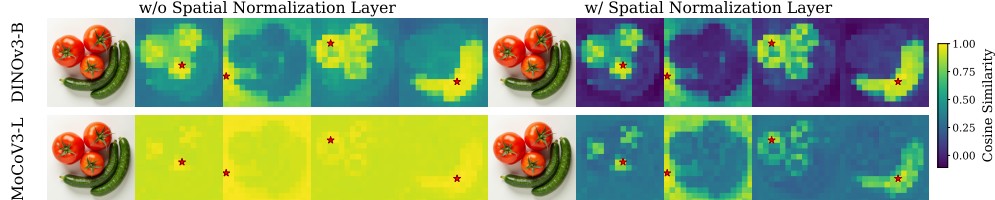

(a) *Simpler projection layer for REPA.* Standard MLP projection layer in REPA (middle) loses spatial information while transferring features from target representation (left) to diffusion features. Instead using a simpler convolution layer leads to better spatial information transfer (right).

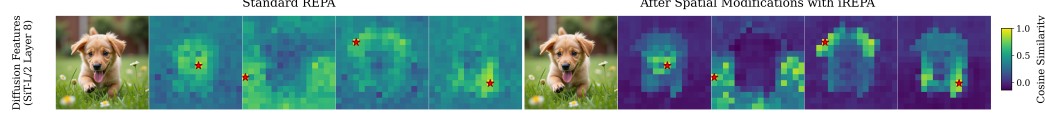

(b) *Spatial normalization layer.* Patch tokens of pretrained vision encoders have a global component which limits spatial contrast. This causes the tokens in one semantic region (*e.g.*, tomato) to show quite decent cosine similarity with unrelated tokens (*e.g.*, background or cucumber). We hypothesize that we can sacrifice this global information to improve the spatial contrast between patch tokens - leading to better generation performance.

(c) *Overall impact* of improved training recipe (iREPA) on spatial structure of diffusion features.

Figure 6: Motivating two straightforward modifications (iREPA) to enhance spatial feature transfer from target representation to diffusion features. All results reported with SiTL/2 with REPA at 100K.

- Can iREPA consistently improve the convergence speed of diffusion transformers over REPA across diverse external representations? (Figure 7, 8, 12, 13, 15)
- Is iREPA scalable in terms of model size and generalize across variations in training settings? (Table 1 [a,b,c], 2, 4, 5, 6, and, Figure 12, 16)
- Does iREPA generalize across more recent representation alignment methods such as REPA-E (Leng et al., 2025a), MeanFlow w/ REPA (Geng et al., 2025)? and pixel-space diffusion models such as JiT w/ REPA (Li & He, 2025b)? (Table 3 [a,b])

**Convergence Speed.** We evaluate the convergence behavior of iREPA across diverse vision encoders (DINOv3-B, WebSSL-1B, PE-Core-G, CLIP-L, MoCov3, PE-Lang-G), and model sizes (SiT-XL/2 and SiT-B/2). Results are shown in Fig. 7. We find that iREPA consistently helps achieve faster convergence over baseline REPA across variations in both target representation and model sizes.

**Target representation.** We analyze the generalization of iREPA across different vision encoders in Fig. 8 and Table 4. We observe that iREPA consistently improves the generation quality across all vision encoders. Additional comparisons across all 27 encoders are provided in Appendix C, E.

| Enc. Size | IS↑ | FID↓ | sFID↓ | Pr.↑ | Rc.↑ |
|---|---|---|---|---|---|
| PE-B (90M) | 59.6 | 22.5 | 6.23 | 0.63 | 0.60 |
| **+iREPA** | **73.3** | **17.5** | **6.04** | **0.66** | **0.62** |
| PE-L (320M) | 48.6 | 28.7 | 6.53 | 0.59 | 0.60 |
| **+iREPA** | **76.3** | **17.6** | **6.36** | **0.65** | **0.61** |
| PE-G (1.88B) | 42.7 | 32.3 | 6.70 | 0.57 | 0.59 |
| **+iREPA** | **70.8** | **19.5** | **6.35** | **0.64** | **0.61** |

| Model Size | IS↑ | FID↓ | sFID↓ | Pr.↑ | Rc.↑ |
|---|---|---|---|---|---|
| SiT-B | 27.5 | 49.50 | 7.00 | 0.46 | 0.59 |
| **+iREPA** | **34.1** | **43.37** | 7.87 | **0.50** | **0.60** |
| SiT-L | 55.7 | 24.10 | 6.25 | 0.62 | 0.60 |
| **+iREPA** | **66.9** | **20.28** | **6.14** | **0.63** | **0.62** |
| SiT-XL | 70.3 | 19.06 | 5.83 | 0.65 | 0.61 |
| **+iREPA** | **77.9** | **16.96** | 6.26 | **0.66** | **0.61** |

| Aln. Depth | IS↑ | FID↓ | sFID↓ | Pr.↑ | Rc.↑ |
|---|---|---|---|---|---|
| Layer 4 | 28.1 | 49.0 | 6.83 | 0.46 | 0.60 |
| **+iREPA** | **41.7** | **36.0** | **6.61** | **0.52** | **0.62** |
| Layer 6 | 26.9 | 50.6 | 7.00 | 0.46 | 0.59 |
| **+iREPA** | **42.4** | **35.3** | **6.90** | **0.53** | **0.62** |
| Layer 8 | 24.7 | 54.8 | 7.35 | 0.44 | 0.58 |
| **+iREPA** | **38.3** | **38.9** | **7.19** | **0.51** | **0.62** |

|  (a) Vision Encoder size | (b) Model size | (c) Encoder depth (SiT-B/2) |
|---|---|---|

Table 1: **Variation in training settings.** We find that iREPA leads to consistent gains over baseline REPA across diverse training settings. Unless otherwise specified all results are reported using Dinov2 as encoder, SiTXL/2, 100K steps and vanilla-REPA as baseline.

**Encoder size.** We analyze the generalization of iREPA across different encoder sizes. Table 1a shows results analyzing generalization of iREPA across different encoder sizes; PE-B (90M), PE-L (320M), PE-G (1.88B). We see that use of iREPA consistently helps improve the performance across

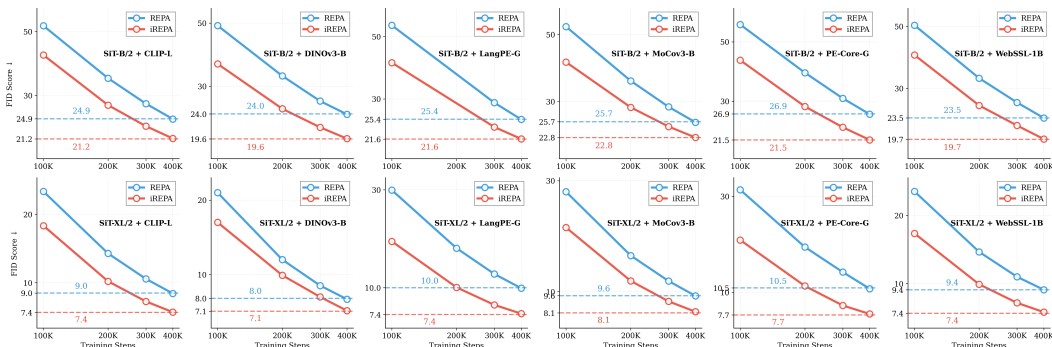

Figure 7: **Accentuating spatial features helps consistently improve convergence speed.**

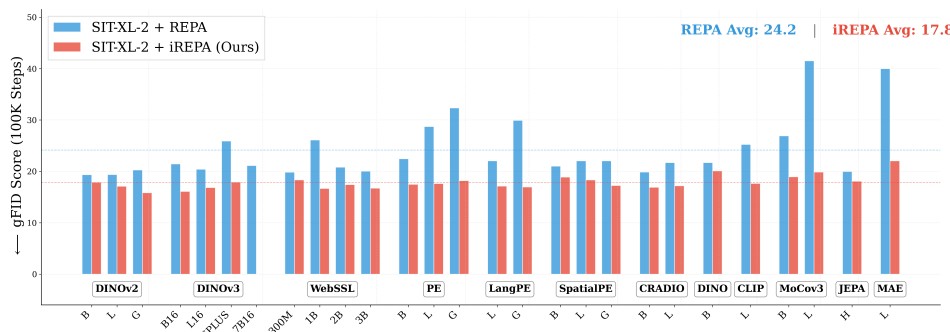

Figure 8: **Variation in target representation**. Across all 27 vision encoders, we find that accentuating transfer of spatial features from target representation to diffusion features (iREPA) leads to consistent improvements in convergence speed. See Appendix C for more results across diverse settings.

all encoder sizes. Interestingly, the percentage improvement also increases with increasing encoder size (22.2% for PE-B, 38.8% for PE-L, 39.6% for PE-G).

**Scalability.** We analyze the scalability of iREPA across different model scales in Table 1b. We observe that the spatial improvements not only consistently improve performance, but larger percentage gains are seen with larger models; showing that spatial improvements are scalable with model size.

**Encoder depth.** Table 1c analyzes generalization of iREPA across different alignment depths. All results are with SiT-B/2 at 100K iterations using DINOv3-B as target representation. We observe consistent improvements over baseline REPA across different alignment depths.

**Abalation on different components.** We also study the role of different components in iREPA in Table 2. We observe that both spatial normalization and convolution projection layer significantly improve the generation quality over baseline REPA; with the best results achieved by using both.

**Training recipe.** Lastly, we analyze the generalization of iREPA across different training recipes such as REPA-E (Leng et al., 2025a) and MeanFlow w/ REPA (Geng et al., 2025). Table 3 shows that spatial improvements with iREPA lead to convergence gains across different training recipes.

**Classifier-free guidance.** We evaluate the generation quality of iREPA with CFG in Table 4. presents results across different vision encoders at 400K training iterations. We find that across different encoders, iREPA leads to faster convergence both with and without classifier-free guidance.

**Pixel-space diffusion (JiT).** We also evaluate iREPA on pixel-space diffusion models such as JiT-B (Li & He, 2025b). Figure 9 shows results across with both REPA and iREPA using JiT (Li & He, 2025b). We observe that accentuating transfer of spatial information consistently achieves faster convergence with REPA across various vision encoders (e.g., DiNOv2, DiNOv3, PE etc.).

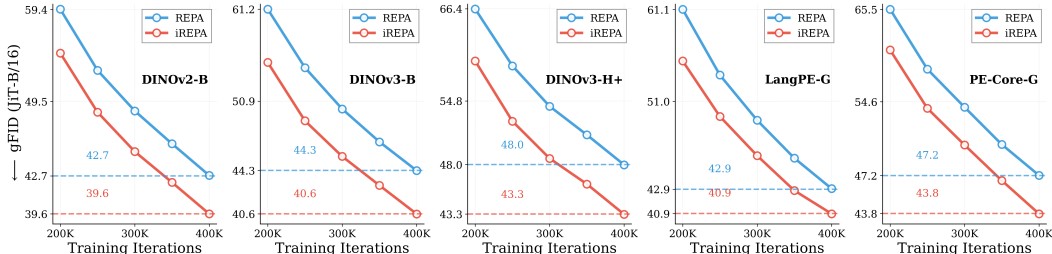

Figure 9: **Convergence comparison with pixel-space diffusion (JiT).** FID convergence curves comparing REPA vs iREPA with pixel space JiT (Li & He, 2025b) across different vision encoders. Accentuating transfer of spatial structure helps consistently improve the convergence speed of across different vision encoders for pixel-space diffusion models. All results are reported with JiTB/16 (Li & He, 2025b), 256 batch size and without classifier-free guidance. Refer Table 10 for further results.

| Method | DINOv2-B | | | DINOv3-B | | | WebSSL-1B | | | PE-Core-G | | |
|---|---|---|---|---|---|---|---|---|---|---|---|---|
| | FID↓ | IS↑ | sFID↓ | FID↓ | IS↑ | sFID↓ | FID↓ | IS↑ | sFID↓ | FID↓ | IS↑ | sFID↓ |
| Baseline REPA | 19.06 | 70.3 | 5.83 | 21.47 | 63.4 | 6.19 | 26.10 | 53.0 | 6.90 | 32.35 | 42.7 | 6.70 |
| iREPA (w/o spatial norm) | 18.52 | 73.3 | 6.11 | 17.76 | 74.7 | 5.81 | 21.17 | 64.6 | 6.27 | 24.97 | 57.4 | 6.21 |
| iREPA (w/o conv proj) | 17.66 | 72.8 | 6.03 | 18.28 | 70.8 | 6.18 | 18.44 | 71.0 | 6.22 | 21.72 | 61.5 | 6.26 |
| **iREPA (full)** | **16.96** | **77.9** | 6.26 | **16.26** | **78.8** | 6.14 | **16.66** | **77.5** | **6.18** | **18.19** | **75.0** | **6.03** |

Table 2: **Ablation on different components.** We see that across different encoders - both spatial normalization layer and convolution projection layer (§4) lead to significant gains in convergence speed; with best results obtained using both. All results reported at 100K steps and SiT-XL/2.

> • **Finding 3.** *Accentuating transfer of spatial structure from target representation to diffusion features helps boost convergence speed with representation alignment (REPA).*

## 5 RELATED WORK

We discuss the most relevant related work here and provide a detailed discussion in Appendix H.

**Representation alignment for generation.** Many recent works explore use of external representations for improving diffusion model training (Pernias et al., 2023; Fuest et al., 2024). Notably, recent works (Yu et al., 2024; Yao & Wang, 2025; Leng et al., 2025a;b; Kouzelis et al., 2025) show that significant performance gains can be achieved by aligning internal diffusion features with clean image features from a pretrained vision encoder. (Zhang et al., 2025; Wu et al., 2025) extend this idea to video generation and 3D generation respectively. (Ma et al., 2025a) shows that representation alignment can be used to improve training of unified models. Despite these emperical successes, there remains limited understanding of the precise mechanisms through which self-supervised features enhance diffusion model training. In this paper, we try to understand what aspect of the target representation matters for generation, and use it to propose an improved training recipe.

**Spatial vs global information tradeoff in pretrained vision encoders.** Recent works explore the tradeoff between global and spatial information in pretrained vision encoders (Bolya et al., 2025; Siméoni et al., 2025). (Siméoni et al., 2025) show that continued training of self-supervised vision representations can lead to increased similarity between global CLS token and patch tokens — leading to worse performance on dense spatial tasks. (Bolya et al., 2025; Heinrich et al., 2024) specifically train spatial-tuned models for dense spatial tasks. In this paper, we show that for generation, spatial structure of a vision encoder matters more then its global information. We hope this motivates future research on better selecting and training external representations for generation.

## 6 CONCLUSION

In this paper, we study what truly drives the effectiveness of representation alignment, global information or the spatial structure of the target representation? Through large-scale empirical analysis

| Encoder | IS↑ | FID↓ | sFID↓ | Prec.↑ | Rec.↑ |
|---|---|---|---|---|---|
| WebSSL-1B | 52.8 | 26.5 | 5.20 | 0.620 | 0.585 |
| **+iREPA-E** | **87.0** | **13.2** | 5.28 | **0.699** | **0.598** |
| PE-G | 50.9 | 25.9 | 5.68 | 0.612 | 0.576 |
| **+iREPA-E** | **80.0** | **16.4** | **5.40** | **0.667** | **0.616** |
| DINOv3-B | 82.2 | 14.4 | 4.68 | 0.694 | 0.596 |
| **+iREPA-E** | **93.6** | **11.7** | **4.57** | **0.703** | **0.613** |
| DINOv2-B | 87.5 | 12.9 | 5.40 | 0.708 | 0.586 |
| **+iREPA-E** | **91.3** | **12.1** | **4.86** | **0.712** | **0.602** |

(a) REPA-E (SiT-XL/2)

| Encoder | w/o CFG | | | | w/ CFG | | | |
|---|---|---|---|---|---|---|---|---|
| | 4 NFE | | 1 NFE | | 4 NFE | | 1 NFE | |
| | IS↑ | FID↓ | IS↑ | FID↓ | IS↑ | FID↓ | IS↑ | FID↓ |
| WebSSL-1B | 27.22 | 51.40 | 24.14 | 58.67 | 87.85 | 16.59 | 69.06 | 23.74 |
| **+iREPA** | **31.48** | **45.67** | **27.33** | **55.70** | **100.74** | **13.89** | **78.67** | **20.69** |
| DINOv3-B | 28.36 | 49.58 | 25.47 | 57.01 | 93.30 | 15.56 | 72.41 | 22.55 |
| **+iREPA** | **33.63** | **44.52** | **29.67** | **53.75** | **124.54** | **11.05** | **98.93** | **17.32** |
| DINOv2-B | 28.77 | 48.87 | 25.43 | 56.63 | 94.40 | 15.28 | 71.95 | 22.17 |
| **+iREPA** | **33.63** | **44.52** | **29.67** | **53.75** | **111.44** | **12.59** | **90.25** | **18.62** |

(b) MeanFlow w/ REPA (SiT-B/2)

Table 3: **Variation in training recipes.** We apply our spatial improvements on top of REPA-E (a) and Meanflow with REPA (b); achieving consistent gains. For Meanflow, we report results at 1 and 4 NFEs. CFG value of 2.0 is used for Meanflow. All results are reported at 100K training iterations.

| Vision Encoder | Steps | w/o CFG | | | | | w/ CFG | | | | |
|---|---|---|---|---|---|---|---|---|---|---|---|
| | | IS↑ | FID↓ | sFID↓ | Prec.↑ | Rec.↑ | IS↑ | FID↓ | sFID↓ | Prec.↑ | Rec.↑ |
| DINOv2-B | 100K | 69.20 | 19.3 | 5.89 | 0.64 | 0.61 | 157.2 | 6.35 | 5.91 | 0.769 | 0.536 |
| **+iREPA** | 100K | **77.92** | **16.9** | 6.26 | **0.66** | **0.61** | **179.3** | **5.15** | 6.23 | **0.783** | **0.544** |
| DINOv2-B | 400K | 127.4 | 7.76 | 5.06 | 0.70 | 0.66 | 263.0 | 1.98 | 4.60 | 0.799 | 0.610 |
| **+iREPA** | 400K | **128.6** | **7.52** | **4.89** | **0.71** | **0.65** | **268.8** | **1.93** | **4.59** | **0.799** | **0.600** |
| DINOv3-B | 100K | 63.64 | 21.4 | 6.14 | 0.63 | 0.60 | 144.0 | 7.57 | 6.09 | 0.762 | 0.526 |
| **+iREPA** | 100K | **78.79** | **16.2** | **6.14** | **0.66** | **0.61** | **181.9** | **4.87** | 6.10 | **0.780** | **0.547** |
| DINOv3-B | 400K | 126.7 | 8.10 | 5.06 | 0.70 | 0.66 | 261.2 | 1.99 | 4.58 | 0.799 | 0.609 |
| **+iREPA** | 400K | **132.9** | **7.13** | **4.93** | **0.71** | **0.66** | **272.4** | **1.89** | **4.58** | **0.799** | **0.600** |
| WebSSL-1B | 100K | 53.87 | 25.5 | 6.57 | 0.61 | 0.59 | 124.0 | 9.59 | 6.37 | 0.756 | 0.515 |
| **+iREPA** | 100K | **77.47** | **16.6** | **6.18** | **0.66** | **0.61** | **177.6** | **5.09** | **6.11** | **0.787** | **0.538** |
| WebSSL-1B | 400K | 116.9 | 9.39 | 5.14 | 0.70 | 0.64 | 250.5 | 2.24 | 4.61 | 0.809 | 0.580 |
| **+iREPA** | 400K | **130.8** | **7.48** | **4.91** | **0.70** | **0.65** | **271.6** | **1.90** | **4.58** | **0.798** | **0.609** |
| PE-Core-G | 100K | 42.74 | 32.3 | 6.70 | 0.57 | 0.59 | 97.2 | 14.1 | 6.56 | 0.714 | 0.525 |
| **+iREPA** | 100K | **75.01** | **18.1** | **6.03** | **0.64** | **0.61** | **176.8** | **5.66** | 6.08 | **0.771** | **0.544** |
| PE-Core-G | 400K | 109.4 | 10.4 | 5.00 | 0.69 | 0.64 | 238.4 | 2.44 | 4.57 | 0.805 | 0.585 |
| **+iREPA** | 400K | **132.0** | **7.78** | 5.02 | **0.70** | **0.65** | **275.4** | **1.93** | 4.59 | **0.796** | **0.606** |

Table 4: **Results across different encoders with and w/o classifier-free guidance**. SiT-XL/2 is used base model. See Appendix E for detailed results on SiT-B/2 & SiT-L/2 across all encoders.

we uncover a surprising finding: spatial structure and not global information, drives the effectiveness of representation alignment. We further study this by introducing two simple modifications which accentuate the transfer of spatial information from target representation to diffusion features. Our simple method, termed iREPA, consistently improves convergence speed with REPA across diverse variations in vision encoders and training recipes. We hope our work will motivate future research to revisit the fundamental working mechanism of representational alignment and how we can better leverage it for improved training of generative models.

## ACKNOWLEDGEMENT

We thank You Jiacheng (@YouJiacheng), Shuming Hu (@ShumingHu), @gallabytes whose comments on X motivated the exploration in this direction (Jiacheng, 2025a;b; Hu, 2025). The authors were glad to find out their original predictions were wrong, which opened the door to new insights.

We also thank Zhengyang Geng for providing the meanflow with REPA implementation and for useful discussion on hyperparameter configuration. We also thank Boyang Zheng, Fred Lu, Nanye (Willis) Ma and Sihyun Yu for insightful discussions and guidance on RAE experiments.

## REPRODUCIBILITY STATEMENT

We provide all implementation details and hyperparameters in Appendix G. We also open-source our code, model checkpoints and analysis results.

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

# A  DETAILED CORRELATION ANALYSIS

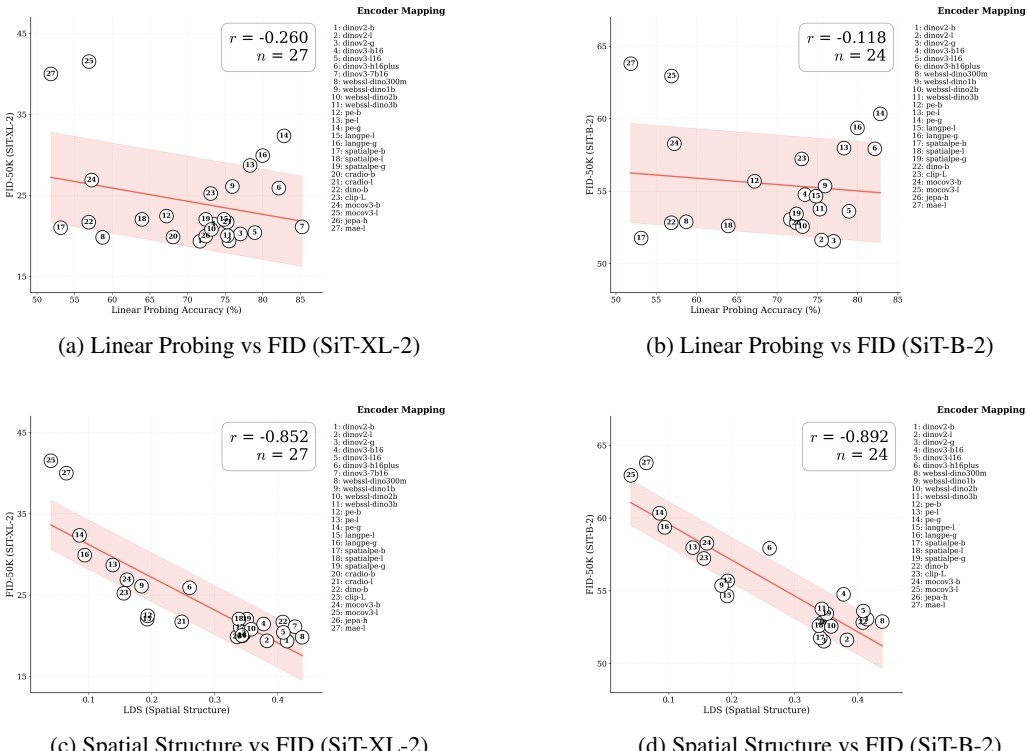

Figure 10: **Detailed correlation analysis with encoder labels.** Top row shows linear probing accuracy vs FID correlation for SiT-XL-2 and SiT-B-2 models. Bottom row shows spatial structure (LDS) vs FID correlation. Each point is labeled with its corresponding encoder number (see legend). The spatial structure metric demonstrates significantly stronger correlation with generation quality across both model scales.

# B SPATIAL SELF-SIMILARITY METRICS

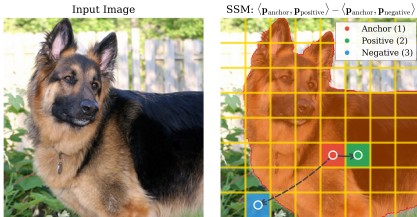

Figure 11: **Explaining semantic region self-similarity (SRSS) metric.** Visual explanation of SRSS metric: For each image, we compute a mask using SAM2 and select anchor-positive-negative triplets. The SSM measures the difference in cosine similarity between anchor-positive pairs (within mask) and anchor-negative pairs (outside mask). Intuitively, larger values indicate that patches on the same semantic region are more similar than patches on unrelated regions — indicating better spatial structure preservation.

**Setup.** Given an image $\mathcal{I}$ and vision encoder $\mathcal{E}$, we extract patch tokens $\mathcal{X} = \mathcal{E}(\mathcal{I}) \in \mathbb{R}^{T \times D}$ with $T = H \times W$ and indices $t \in [T]$ placed on an $H \times W$ lattice. Let $d : [T] \times [T] \to \mathbb{N}$ be the Manhattan distance, and let the (cosine) self-similarity kernel be

$$K_{\mathcal{X}}(t, t') = \frac{\langle \mathbf{x}_t, \mathbf{x}_{t'} \rangle}{\|\mathbf{x}_t\|_2 \|\mathbf{x}_{t'}\|_2} \in [-1, 1],$$

a standard local self-similarity measure (Shechtman & Irani, 2007).

- *Patch token representation:* $\mathcal{X} = \mathcal{E}(\mathcal{I}) \in \mathbb{R}^{T \times D}$ where $T = H \times W$ is the spatial grid of patches (patch index set $[T]$). Also let $d : [T] \times [T] \to \mathbb{N}$ be Manhattan distance on the $H \times W$ grid.
- *Self-similarity kernel:* $K : [T] \times [T] \to \mathbb{R}$ measuring self-similarity (Shechtman & Irani, 2007) between patch tokens. We use the cosine kernel $K_{\mathcal{X}}(t, t') = \langle \mathbf{x}_t, \mathbf{x}_{t'} \rangle / (\|\mathbf{x}_t\|_2 \|\mathbf{x}_{t'}\|_2)$.
- *Spatial self-similarity metric:* a functional $m : \mathcal{K} \times \mathcal{D} \to \mathbb{R}$ measures how self-similarity $K$ between patch tokens varies with lattice distance $d$. Intuitively, larger values indicate stronger spatial organization (near patches more similar to each other than far away patches).

We next discuss several straightforward metrics for measuring the spatial self-similarity structure of the target representations.

**Local vs. Distant Similarity (LDS).** We first consider a simple correlogram contrast (local vs. distant) metric (Huang et al., 1997):

$$\text{LDS}(\mathcal{X}) = \mathbb{E}[K_{\mathcal{X}}(t, t') \,|\, d(t, t') < r_{\text{near}}] \; - \; \mathbb{E}[K_{\mathcal{X}}(t, t') \,|\, d(t, t') \geq r_{\text{far}}].$$

where $r_{\text{near}}$ and $r_{\text{far}}$ are the hyperparameters. By default we use $r_{\text{near}} = r_{\text{far}} = H/2$. We found correlation to be robust to their exact choices. Intuitively, larger values indicate that *on average*, patches that are closer to each other are more similar than patches that are further away.

**Correlation Decay Slope (CDS).** Given the patch tokens $\mathcal{X} = \mathcal{E}(\mathcal{I}) \in \mathbb{R}^{T \times D}$, we compute the spatial correlogram $g_{\mathcal{X}}(\delta) = \mathbb{E}[K_{\mathcal{X}}(t, t') | d(t, t') = \delta]$ for distances $\delta \in \{1, \ldots, \Delta\}$. We then fit a least-squares line $\widehat{g}_{\mathcal{X}}(\delta) \approx \alpha + \beta\delta$ and define:

$$\text{CDS}(\mathcal{X}) = -\hat{\beta}$$

where $\hat{\beta}$ is the fitted slope. Larger CDS values indicate faster similarity decay with distance, suggesting stronger spatial organization.

**Semantic-Region Self-Similarity (SRSS).** For each image, we first sample a binary mask $\mathbf{M} \in \mathbb{R}^{H \times W}$ containing a random object using SAM2 (Ravi et al., 2024) and downsampled to $H \times W$. We then select three types of points: anchor, positive, and negative. The anchor and positive points are sampled from within the mask (should have similar semantics), while the negative point is sampled from outside the mask (less related to anchor). Conceptually, if the encoder feature representation preserves the spatial structure, the anchor and positive points should have higher cosine similarity,

while the anchor and negative points should have lower cosine similarity. Thus, we define the Spatial Structure Metric (SSM) as:

$$\text{SSM}(\mathbf{P}) = \mathbb{E}_{\text{anchor} \in \mathbf{M}} \left[ \cos(\mathbf{p}_{\text{anchor}}, \mathbf{p}_{\text{pos}}) - \cos(\mathbf{p}_{\text{anchor}}, \mathbf{p}_{\text{neg}}) \right]$$

where positive points are within Manhattan distance $d \leq r_{near}$ from the anchor within the mask, and negative points are at distance $d \geq r_{far}$ outside the mask.

**RMS Spatial Contrast (RMSC).** Finally, we consider a simple contrast metric to capture the spatial contrast between patch token representations. Given normalized features $\hat{\mathbf{x}}_t = \mathbf{x}_t / \|\mathbf{x}_t\|_2$ for each patch $t$, we compute:

$$\text{RMSC}(\mathcal{X}) = \sqrt{\frac{1}{T} \sum_{t=1}^{T} \|\hat{\mathbf{x}}_t - \bar{\mathbf{x}}\|_2^2}$$

where $\bar{\mathbf{x}} = \frac{1}{T} \sum_{t=1}^{T} \hat{\mathbf{x}}_t$ is the mean of normalized features across all spatial locations. Higher RMSC values indicate greater spatial diversity in the feature representations, suggesting preserved spatial structure, whereas lower values indicate more uniform feature distributions indicating a loss of spatial structure (typically happens global information dimnishes spatial structure (Siméoni et al., 2025)).

## C  VISION ENCODER VARIATION

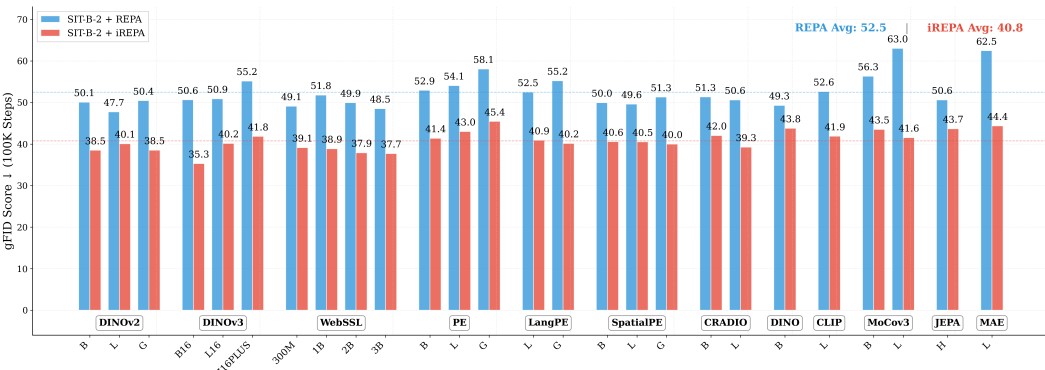

(a) SiT-B-2 (130M parameters)

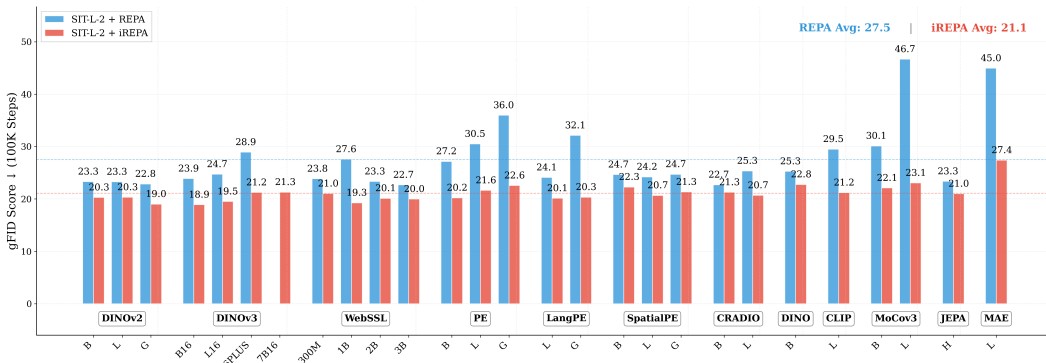

(b) SiT-L-2 (458M parameters)

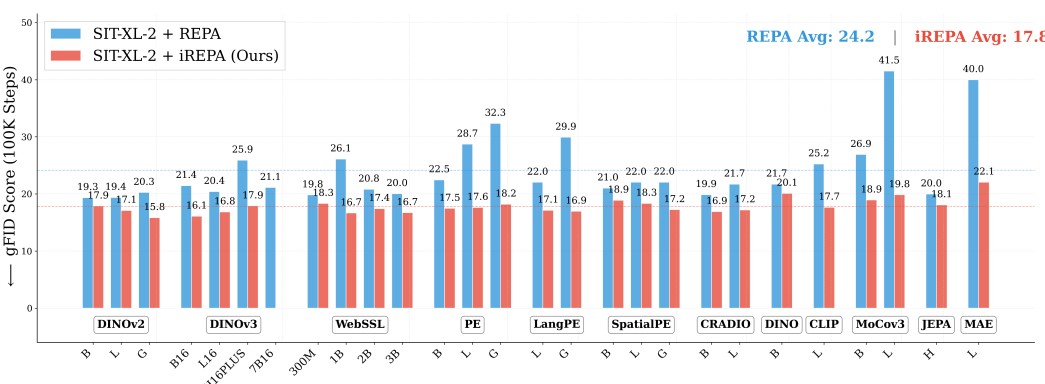

(c) SiT-XL-2 (675M parameters)

Figure 12: **Variation in vision encoders.** Ablation studies showing iREPA improvements across different vision encoders for SiT-B-2, SiT-L-2, and SiT-XL-2 models. iREPA consistently improves generation quality across all encoders and model sizes.

# D    ENCODER DEPTH VARIATION

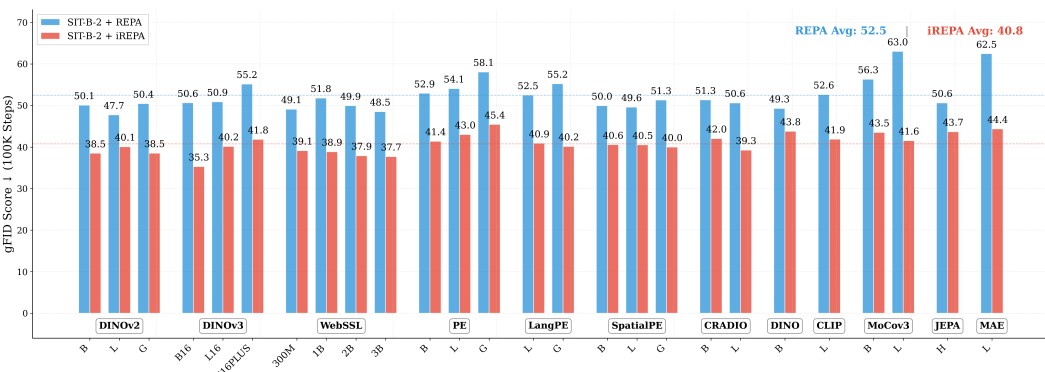

(a) SiT-B-2 with 6 encoder layers

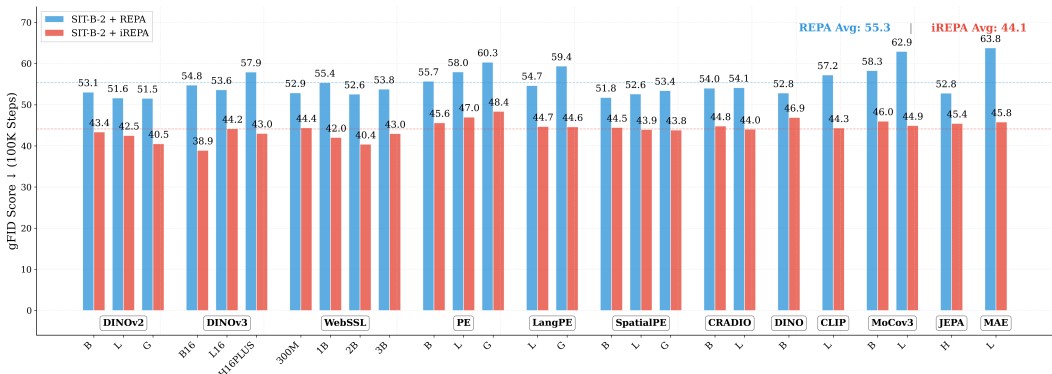

(b) SiT-B-2 with 8 encoder layers

Figure 13: **Effect of encoder depth on iREPA performance.** Comparison of SiT-B-2 performance with different encoder depths (6 vs 8 layers) across various vision encoders. The results show consistent improvements with iREPA regardless of encoder depth, with slightly better performance observed with 8 encoder layers. Both configurations demonstrate significant FID improvements when spatial regularization is applied.

# E   COMPREHENSIVE RESULTS ACROSS VISION ENCODERS

## E.1   SIT-B/2 RESULTS

| Vision Encoder | IS↑ | FID↓ | sFID↓ | Prec.↑ | Rec.↑ |
|---|---|---|---|---|---|
| CLIP-ViT-L | 59.44 | 24.84 | 6.46 | 0.584 | 0.645 |
| **+iREPA** | **69.70** | **21.26** | 6.64 | **0.601** | **0.644** |
| DINOv2-B | 64.92 | 22.75 | 6.54 | 0.593 | 0.653 |
| **+iREPA** | **70.88** | **21.40** | 6.77 | **0.597** | **0.653** |
| DINOv2-L | 66.65 | 22.46 | 6.53 | 0.594 | 0.650 |
| **+iREPA** | **72.41** | **20.99** | 6.84 | **0.600** | **0.650** |
| DINOv3-B16 | 62.57 | 23.91 | 6.77 | 0.584 | 0.647 |
| **+iREPA** | **75.48** | **19.65** | **6.68** | **0.606** | **0.649** |
| DINOv3-H16+ | 61.91 | 24.82 | 6.57 | 0.579 | 0.645 |
| **+iREPA** | **71.76** | **21.80** | 6.84 | **0.592** | **0.655** |
| DINO-B | 58.37 | 24.71 | 6.32 | 0.586 | 0.635 |
| **+iREPA** | **57.55** | **24.41** | 6.43 | **0.587** | **0.638** |
| LangPE-G | 58.42 | 25.37 | 6.32 | 0.583 | 0.641 |
| **+iREPA** | **66.88** | **21.62** | 6.65 | **0.604** | **0.649** |
| MoCov3-B | 57.61 | 25.60 | 6.25 | 0.579 | 0.640 |
| **+iREPA** | **61.41** | **22.80** | 6.46 | **0.596** | **0.639** |
| PE-B | 58.86 | 25.78 | 6.44 | 0.58 | 0.64 |
| **+iREPA** | **69.50** | **21.15** | 6.66 | **0.601** | **0.659** |
| PE-G | 56.09 | 26.84 | 6.45 | 0.568 | 0.641 |
| **+iREPA** | **70.43** | **21.52** | 6.58 | **0.598** | **0.651** |
| SpatialPE-B | 59.19 | 24.28 | 6.37 | 0.586 | 0.646 |
| **+iREPA** | **61.50** | **22.71** | 6.41 | **0.604** | **0.644** |
| SpatialPE-L | 62.13 | 23.87 | 6.51 | 0.59 | 0.65 |
| **+iREPA** | **68.83** | **21.07** | 6.59 | **0.607** | **0.651** |
| WebSSL-1B | 63.20 | 23.48 | 6.33 | 0.593 | 0.646 |
| **+iREPA** | **73.54** | **19.74** | 6.63 | **0.612** | **0.646** |
| WebSSL-2B | 66.56 | 22.27 | 6.69 | 0.596 | 0.654 |
| **+iREPA** | **72.13** | **20.47** | 6.89 | **0.609** | **0.648** |
| WebSSL-300M | 62.06 | 23.64 | 6.48 | 0.590 | 0.646 |
| **+iREPA** | **66.83** | **21.46** | 6.71 | **0.606** | **0.653** |
| WebSSL-3B | 65.91 | 22.55 | 6.53 | 0.598 | 0.654 |
| **+iREPA** | **71.98** | **20.64** | 6.71 | **0.607** | **0.650** |

Table 5: **SiT-B/2 performance across vision encoders at 400K iterations.** iREPA consistently improves generation quality. All baselines are reported using vanilla-REPA (Yu et al., 2024) for training.

## E.2 SiT-L/2 Results

| Vision Encoder | IS↑ | FID↓ | sFID↓ | Prec.↑ | Rec.↑ |
|---|---|---|---|---|---|
| CLIP-ViT-L | 107.7 | 10.6 | 5.20 | 0.682 | 0.643 |
| **+iREPA** | **115.8** | **9.41** | **5.15** | **0.689** | **0.653** |
| DINOv2-B | 113.3 | 9.61 | 5.11 | 0.684 | 0.653 |
| **+iREPA** | **113.9** | **9.36** | **5.11** | **0.690** | **0.655** |
| DINOv3-B16 | 115.3 | 9.41 | 5.24 | 0.687 | 0.651 |
| **+iREPA** | **118.1** | **9.06** | **5.25** | **0.685** | **0.650** |
| DINOv3-H16+ | 110.7 | 10.6 | 5.32 | 0.677 | 0.652 |
| **+iREPA** | **120.5** | **9.64** | **5.29** | **0.681** | **0.663** |
| LangPE-G | 102.8 | 11.2 | 5.13 | 0.684 | 0.637 |
| **+iREPA** | **112.7** | **9.37** | **5.02** | **0.690** | **0.653** |
| MoCov3-B | 102.1 | 11.0 | 5.14 | 0.685 | 0.635 |
| **+iREPA** | **104.7** | **10.1** | **5.00** | **0.690** | **0.641** |
| PE-G | 98.38 | 12.6 | 5.31 | 0.670 | 0.646 |
| **+iREPA** | **117.1** | **9.65** | **5.33** | **0.683** | **0.651** |
| WebSSL-1B | 107.9 | 10.5 | 5.13 | 0.686 | 0.644 |
| **+iREPA** | **121.5** | **8.62** | **5.01** | **0.696** | **0.653** |

Table 6: **SiT-L/2 performance across vision encoders at 400K iterations.** iREPA consistently improves generation quality across all encoders, with particularly strong gains in FID and IS metrics. All baselines are reported using vanilla-REPA (Yu et al., 2024) for training.

# F  ADDITIONAL EXPERIMENTAL RESULTS

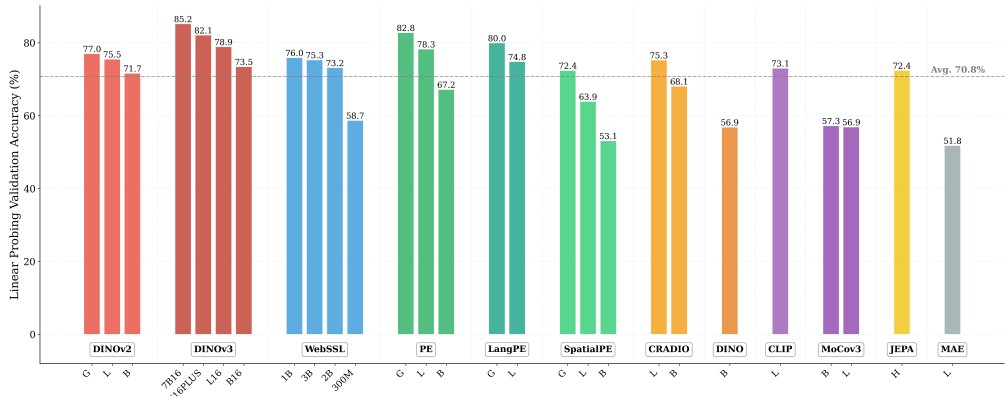

Figure 14: **Global information in mean patch tokens.** We find that in addition to [CLS] token, mean of patch tokens also contains substantial global semantic information. While this helps improve global performance, it reduces the contrast between individual patch tokens, potentially hindering spatial structure transfer. We find that we can remove some of this global information (through mean of patch tokens) to improve spatial structure transfer during representation alignment (§4).

| Vision Encoder | IS↑ | FID↓ | sFID↓ | Prec.↑ | Rec.↑ |
|---|---|---|---|---|---|
| DINOv2-B | 262.97 | 1.98 | 4.60 | 0.799 | 0.610 |
| **+iREPA (Ours)** | **268.79** | **1.93** | **4.59** | **0.799** | **0.600** |
| DINOv3-B | 261.18 | 1.99 | 4.58 | 0.799 | 0.609 |
| **+iREPA (Ours)** | **272.41** | **1.89** | **4.58** | **0.799** | **0.600** |
| WebSSL-1B | 250.53 | 2.24 | 4.61 | 0.809 | 0.580 |
| **+iREPA (Ours)** | **271.59** | **1.90** | **4.58** | **0.798** | **0.609** |
| PE-G | 238.37 | 2.44 | 4.57 | 0.805 | 0.585 |
| **+iREPA (Ours)** | **275.36** | **1.93** | **4.59** | **0.796** | **0.606** |

Table 7: **Generation quality with classifier-free guidance.** Comparison of REPA vs iREPA with CFG (scale 2.0) across different vision encoders. All experiments use SiT-XL/2 trained for 400K iterations with 250 sampling steps. iREPA consistently improves both IS and FID metrics across all encoders.

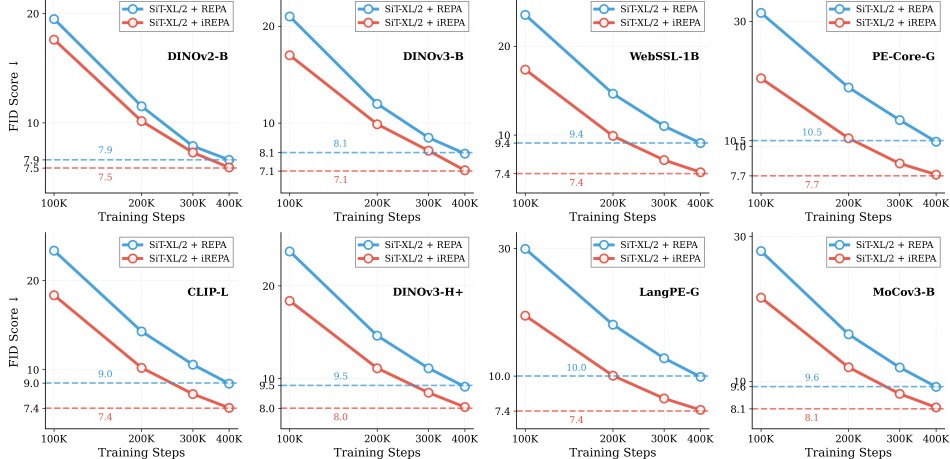

Figure 15: **Accentuating spatial features helps consistently improve convergence speed.** Results for SiT-XL/2 with REPA and iREPA.

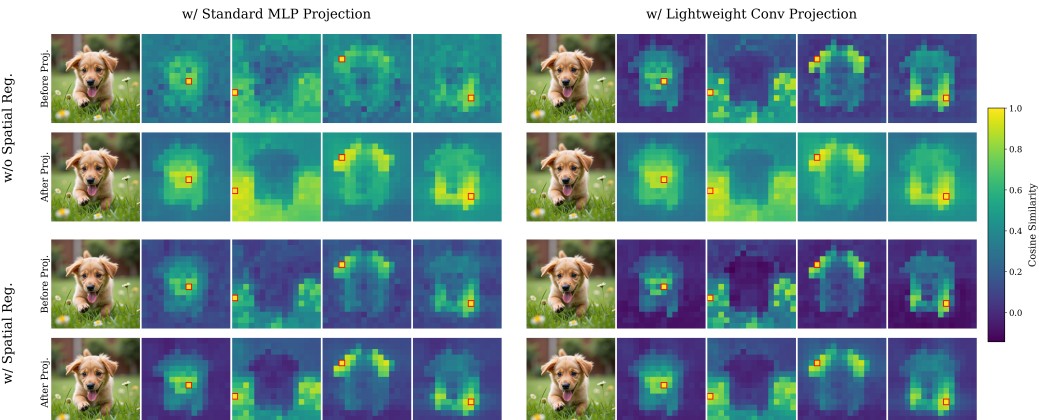

Figure 16: **Visualizing the impact of two straightforward improvements to enhance spatial feature transfer. First,** we find that standard MLP projection layer (top-left) losses spatial information while transferring features from the pretrained encoder (after projection) to diffusion model (before projection). Instead using a simpler convolution layer better preserves the spatial information transfer (top-right). **Second,** we observe that vision encoder features often have limited spatial contrast (§2. This causes the tokens in one semantic region (*e.g.*, dog) to show quite decent cosine similarity with unrelated tokens (*e.g.*, background). We address this with a simple spatial regularization layer, which accentuates the spatial contrast in the learned representation — leading to better generation performance. Best results are obtained while using both (refer Table 2).

# G    IMPLEMENTATION DETAILS

**Training setup.** We follow the same setup as in REPA (Yu et al., 2024) unless otherwise specified. All training is conducted on the ImageNet (Deng et al., 2009) training split. For preprocessing, we adopt the protocol from ADM (Dhariwal & Nichol, 2021), where each image is center-cropped and resized to $256 \times 256$ resolution. We use `stabilityai/sd-vae-ft-mse` VAE (AI, n.d.) throughout our diffusion training and inference. For spatial normalization layer we use $\gamma \in [0.6, 0.8]$ and for projection layer we use a convolutional layer with kernel size 3 and padding 1. For optimization, we use AdamW (Kingma & Ba, 2014; Loshchilov, 2017) with a constant learning rate of $1 \times 10^{-4}$, and a global batch size of 256. During training, we use `bfloat16` mixed precision and `torch.compile` to accelerate training, and gradient clipping and exponential moving average (EMA) to the generative model for stable optimization.

For REPA-E (Leng et al., 2025a) and JiT (Li & He, 2025a) experiments, we use the official open-source implementation. For MeanFlow (Geng et al., 2025) experiments, we received the implementation (including REPA) from original authors and adapt it to introduce the two straightforward changes for iREPA (§4).

**Evaluation.** For image generation evaluation, we strictly follow the ADM setup (Dhariwal & Nichol, 2021). We report generation quality using Fréchet inception distance (gFID) (Heusel et al., 2017), structural FID (sFID) (Nash et al., 2021), inception score (IS) (Salimans et al., 2016), precision (Prec.) and recall (Rec.) (Kynkäänniemi et al., 2019), measured on 50K generated images. For sampling, we follow the approach in REPA (Yu et al., 2024), using the SDE Euler-Maruyama sampler with 250 steps. For JiT (Li & He, 2025a), we use 50 inference steps following official implementation.

# H    MORE DISCUSSION ON RELATED WORK

**Classical spatial features in computer vision.** Spatial feature extraction has long been fundamental to computer vision. Classical methods like SIFT (Lowe, 1999), HOG (Dalal & Triggs, 2005), SURF (Bay et al., 2006), and ORB (Rublee et al., 2011) providing robust local descriptors for tasks ranging from object detection to image matching. While these handcrafted features excel at capturing local spatial patterns and geometric invariances, recent work in generative modeling has primarily focused on representations from modern self-supervised methods that demonstrate strong global classification performance, such as DINOv2 (Oquab et al., 2024) and CLIP (Radford et al., 2021). Our findings suggest a different perspective: since spatial structure preservation is critical for generation quality, even classical spatial features could potentially improve diffusion training when properly aligned. This highlights the potential of leveraging the full spectrum of spatial feature extractors, from traditional handcrafted descriptors to modern learned representations, provided that they maintain strong spatial coherence.

**Pretrained visual encoders for generative models.** Pretrained visual encoders have supported generative models in several capacities: as discriminators to accelerate GAN convergence (Goodfellow et al., 2020; Sauer et al., 2021; 2022; 2023; Radford et al., 2021), as teachers in adversarial distillation for diffusion models (Sauer et al., 2024; Kang et al., 2024), and more recently as alignment targets. In GANs (Goodfellow et al., 2020), pretrained features have not only improved convergence speed but also enabled scaling to large datasets, as demonstrated by StyleGAN-XL (Sauer et al., 2022) and StyleGAN-T (Sauer et al., 2023) with CLIP (Radford et al., 2021) features. For diffusion models, adversarial distillation leverages pretrained encoders to guide student networks toward higher-fidelity samples, showing clear improvements in perceptual quality. In particular, REPA (Yu et al., 2024) aligns diffusion features with external encoders, demonstrating that representation alignment can improve both generation convergence and quality. Building on this direction, we focus not only on alignment but specifically on spatial structure perseverance rather than their discriminative capabilities.

**Denoising transformers.** Transformer architectures have become the dominant backbone for scalable generative modeling, with various formulations including diffusion transformers (Peebles & Xie, 2023) and flow matching variants (Ma et al., 2024). Recent architectures like GenTron (Chen et al., 2023) scale transformers to over 3B parameters for text-to-image synthesis. U-ViT (Bao et al., 2023) demonstrates that pure transformer backbones without U-Net inductive biases can achieve competitive performance. ARDiT (Tang et al., 2024) and DART (Jing et al., 2024) explore

autoregressive formulations that combine denoising with sequential generation, enabling flexible trade-offs between quality and speed. Despite these architectural advances, training these models from scratch remains computationally expensive, often requiring millions of iterations to achieve good generation quality. While representation alignment methods like REPA (Yu et al., 2024) have shown that pretrained features can dramatically accelerate convergence, the mechanisms behind this improvement remained unclear. Our analysis reveals that the key benefit comes from preserving spatial structure rather than semantic alignment, explaining why certain encoders provide stronger acceleration than others and guiding the design of more effective alignment strategies. While (Wang et al., 2025c) propose improvements to REPA training through early stopping, they hypothesize that "REPA predominantly distills global semantic information while leaving structural information untapped." In contrast, our analysis reveals that spatial structure (not global semantic information) already plays a very significant role in the effectiveness of REPA (§2, §3). Thus, while we indeed find that spatial structure remains underexploited (§4) in REPA, we surprisingly find that majority of improvements in REPA are already coming from introducing an inductive bias for spatial structure (not global information).

**Denoising as self-supervised learning task.** The connection between denoising and representation learning has been explored from multiple perspectives. Early work by Abstreiter et al. (2021) extended diffusion objectives for representation learning, demonstrating that denoising naturally learns meaningful features. SODA (Hudson et al., 2024) introduces a diffusion model with an information bottleneck to learn compact representations. Chen et al. (2024) deconstruct diffusion models and find that a simple denoising autoencoder suffices for strong self-supervised performance. Wang & He (2025) introduce a dispersive loss to encourage internal representation diversity in diffusion models, improving generative performance without external encoders. Similarly, Jiang et al. (2025) propose Self-Representation Alignment (SRA) to align a diffusion transformer's latent features across noise levels, providing self-guidance without an auxiliary model. These works establish denoising as a fundamental self-supervised task that naturally encourages learning of robust features. Our findings complement this view by showing that when diffusion models are aligned with strong spatial representations from self-supervised encoders, both the generative and discriminative capabilities improve, suggesting that spatial structure preservation is a key factor in this synergy.

**Scaling self-supervised vision encoders.** Recent years have seen remarkable progress in scaling self-supervised vision models to unprecedented sizes and datasets. DINOv3 (Siméoni et al., 2025) trains a 7B parameter ViT on 1.7 billion images without labels by aligning representations from different augmentations. WebSSL (Fan et al., 2025) demonstrates that visual models trained on more than 2 billion images can match language-supervised models like CLIP (Radford et al., 2021) on vision-language tasks without language supervision. C-RADIO (Heinrich et al., 2024) combines multiple teacher models through distillation to create versatile encoders that excel across diverse visual domains. I-JEPA (Assran et al., 2023) explores predictive architectures that learn by predicting masked regions in representation space, instead of reconstructing pixels directly. SAM (Ravi et al., 2024) specializes in promptable segmentation through large-scale supervised training. The Perception Encoder family (Bolya et al., 2025; Cho et al., 2025) shows that intermediate features often outperform final representations for dense prediction tasks. While these models are typically evaluated on global tasks like image classification, our work reveals a crucial insight: strong performance on discriminative benchmarks does not necessarily translate to better generation quality. Instead, we find that encoders preserving spatial structure, regardless of their classification accuracy—provide the most benefit for diffusion training. This suggests that the evaluation metrics for self-supervised encoders should be reconsidered when targeting generative applications, with spatial coherence being as important as semantic understanding.

# I NOTE ON LLM USAGE

We use GPT-5 (OpenAI, 2025) for considering different variations of the spatial structure metrics discussed in the paper §3. Furthermore, all figures in the paper are directly generated from our experiment logs and checkpoints using Claude-Code (Anthropic, 2025). Additionally we use LLM help for searching and formulating relevant work in Appendix H. We use cursor in some parts to help with paper writing.

## J  ADDITIONAL DISCUSSION ON SPATIAL STRUCTURE METRICS

**Spatial structure metrics can be used to measure the representation gap.** Yu et al. (2024) use linear probing to measure the representation gap, using the increase in validation accuracy to explain the effectiveness of REPA. We find that spatial structure metrics (SSM) can also be used to measure the representaiton gap and explain the improvements across models. As shown in Figure 17, we see that representation alignment helps close the gap between SSM performance of a pretrained encoder like DINOv2 and the diffusion features.

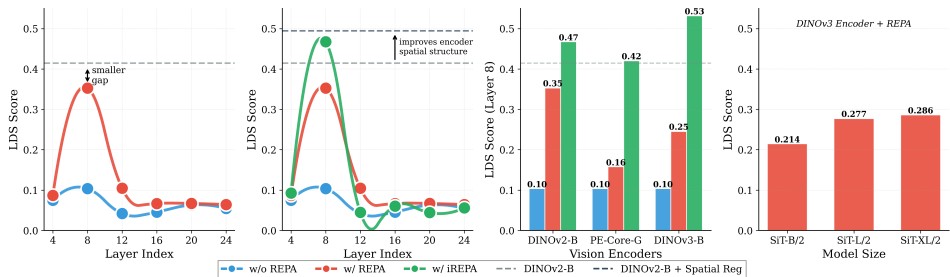

Figure 17: **Representation alignment as bridging the spatial feature gap. a)** We find that representation alignment can also be seen as bridging the spatial feature gap between diffusion and vision encoder patch features. **b)** iREPA (§4) accentuates the spatial features of the vision encoder (at cost of some global information) — helping achieve better generation performance. **c)** iREPA helps consistently improve performance across different vision encoders. **d)** Spatial structure improvements scale with model size. All results are reported with SiT-XL/2 at 400k iterations.

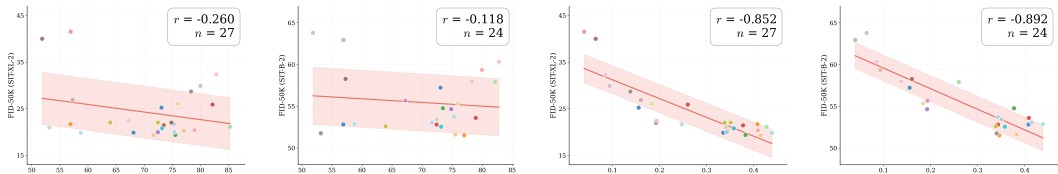

Figure 18: **Spatial structure better correlates with generation quality than linear probing.** Correlation analysis across 27 vision encoders. **Left two:** Linear probing accuracy vs FID for SiT-XL-2 (Pearson $r = -0.26$) and SiT-B-2 ($r = -0.12$) shows weak correlation. **Right two:** Spatial structure (LDS) vs FID for SiT-XL-2 ($r = -0.85$) and SiT-B-2 ($r = -0.89$) shows strong correlation. See Figure 10 in Appendix for detailed plots with encoder labels.

# K  ADDITIONAL ANECDOTAL COMPARISONS

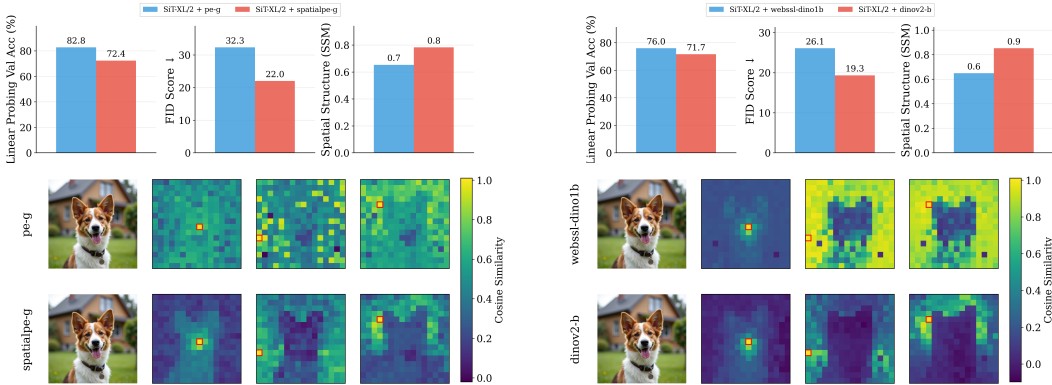

Figure 19: **Motivating anecdotes from recent SSL encoders — spatial structure matters. Top row:** Metrics comparison showing inverse relationship between ImageNet accuracy and generation quality. **Left:** PE-g achieves higher accuracy (82.8% vs 72.4%) but worse FID (32.3 vs 22.0) than Spatial-PE-g, with much lower spatial structure (LDS: 0.1 vs 0.4). **Right:** WebSSL-dino1b shows higher accuracy (76.0% vs 71.7%) but worse FID (26.1 vs 19.3) than DINOv2-b, with weaker spatial structure (LDS: 0.2 vs 0.4). **Bottom row:** Spatial cosine similarity visualizations confirm that encoders with better generation (Spatial-PE-g, DINOv2-b) maintain clear spatial coherence, while those optimized for classification (PE-g, WebSSL) lose spatial structure.

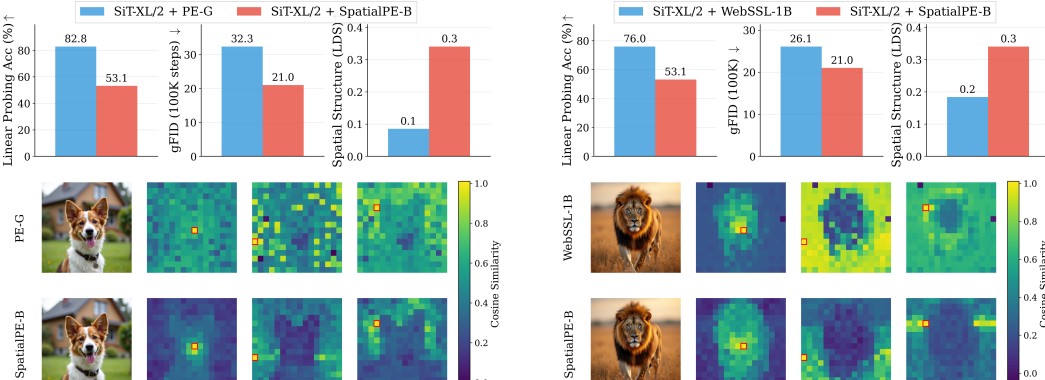

Figure 20: **Motivating examples — spatial structure matters. Top:** Metrics comparison showing inverse relationship between ImageNet accuracy and generation quality. *Left:* $PE_{core}$-G, despite having significantly higher validation accuracy (82.8% vs. 53.1%), shows worse generation quality compared to $PE_{Spatial}$-B (Bolya et al., 2025). *Right:* Similarly, WebSSL-1B (Fan et al., 2025) also shows much better global performance (76.0% vs. 53.1%), but worse generation. **Bottom:** We find that spatial structure instead provides a better predictor of generation quality than global performance. See §3 for spatial structure metric. All results reported at 100K using SiT-XL/2 and REPA.

## L    FURTHER DISCUSSION

**Role of Spatial Normalization** We visualize the impact of spatial normalization on vision encoder representations. Vision encoders can have significant global components or overlays that limit the contrast between spatial tokens. For example, tokens in one semantic region can be highly correlated with tokens in unrelated regions (e.g., background), reducing the spatial distinctiveness of features. Spatial normalization removes this global overlay to enhance contrast between different spatial tokens, allowing the model to better preserve local spatial structure while reducing the dominance of global information that can interfere with generation quality.

The following figures demonstrate this effect across different examples, showing how spatial normalization transforms the feature representations to emphasize spatial structure:

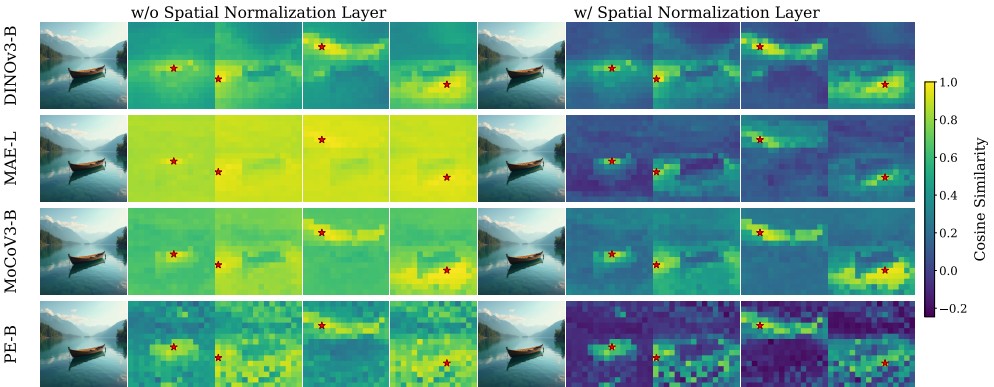

Figure 21: Visualizing impact of spatial normalization (Example 1). The heatmaps show token similarity patterns before and after spatial normalization. Without normalization (left), global components create high correlations across unrelated regions. With spatial normalization (right), local spatial structure is enhanced while reducing global interference, resulting in more distinct semantic boundaries.

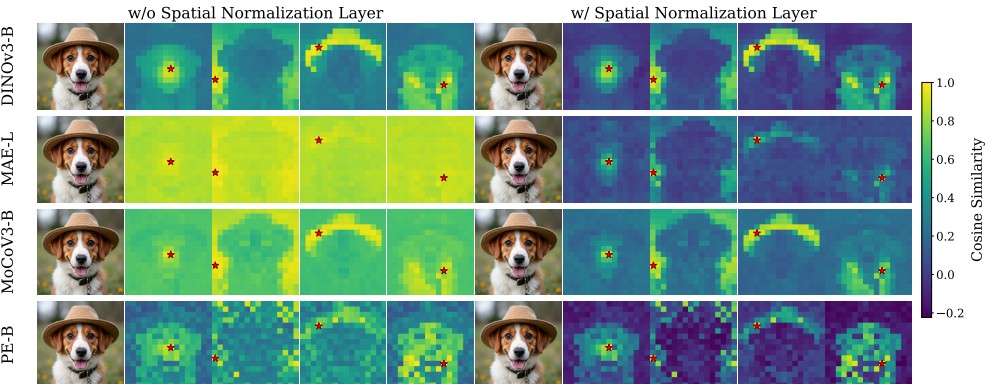

Figure 22: Visualizing impact of spatial normalization (Example 3). The heatmaps show token similarity patterns before and after spatial normalization. Without normalization (left), global components create high correlations across unrelated regions. With spatial normalization (right), local spatial structure is enhanced while reducing global interference, resulting in more distinct semantic boundaries.

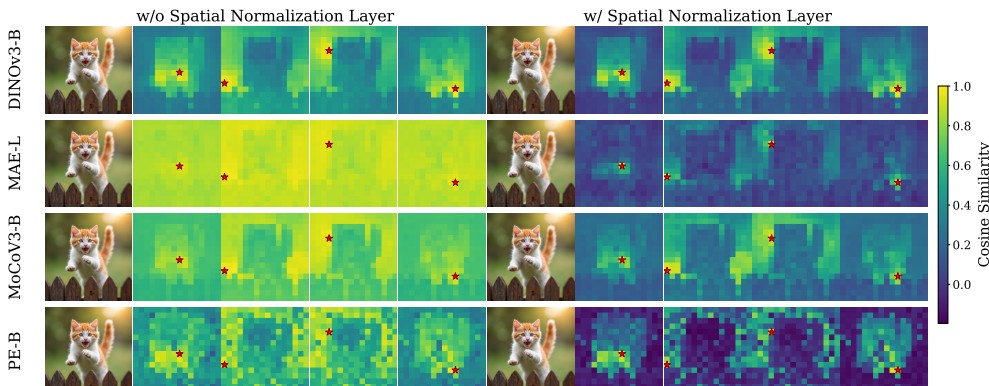

Figure 23: Visualizing impact of spatial normalization (Example 2). Feature similarity maps demonstrate how spatial normalization improves spatial contrast. The original features (left) exhibit a global overlay that reduces distinction between foreground and background regions. After normalization (right), spatial tokens become more locally coherent, with clearer separation between different semantic regions.

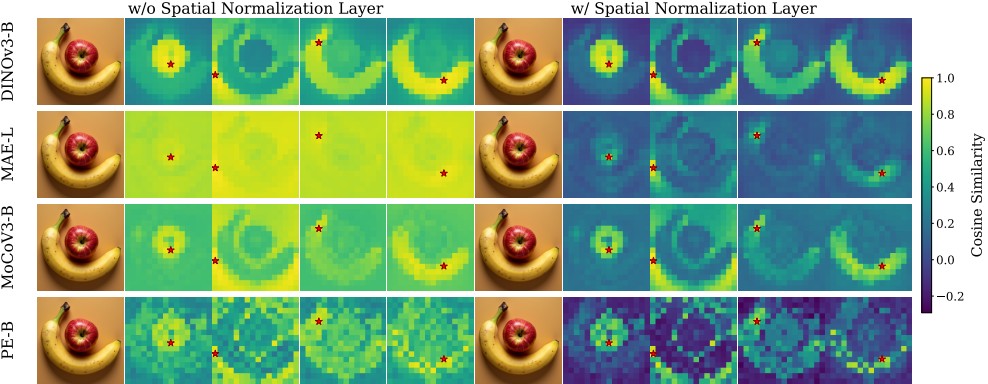

Figure 24: **Visualizing impact of spatial normalization** (Example 4). The heatmaps show token similarity patterns before and after spatial normalization. Without normalization (left), global components create high correlations across unrelated regions. With spatial normalization (right), local spatial structure is enhanced while reducing global interference, resulting in more distinct semantic boundaries.

# M    ADDITIONAL RESULTS

## M.1    ADDITIONAL RESULTS AT HIGHER RESOLUTIONS

**Setup.**    We conduct further experiments on ImageNet-512 (Deng et al., 2009) to evaluate the generalization of thr proposed spatial improvements at higher resolutions. We follow the same setup as REPA (Yu et al., 2024), and report results across different choice of pretrained encoders (DINOv2, DINOv3, WebSSL, PE, etc.). All results are reported using SiT-XL and SiT-B using 50 NFE at inference w/o classifier free guidance. Fig. 25 shows the results with REPA before and after application of spatial improvements (iREPA). We observe that the spatial improvements also generalize to higher resolutions. Furthermore, consistent gains are observed across different choice of pretrained encoders (DINOv2, DINOv3, WebSSL, PE, etc.).

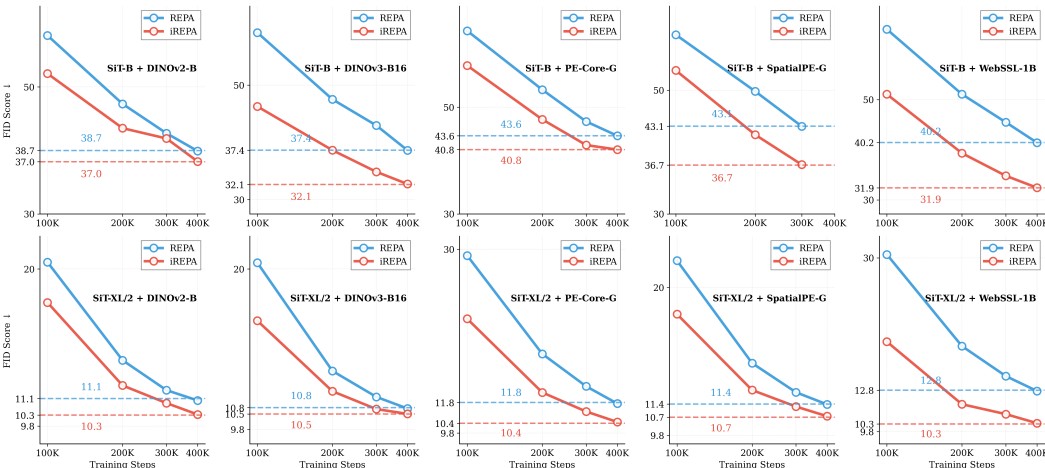

Figure 25: **Convergence results at 512x512 resolution**. Accentuating spatial features helps consistently improve convergence speed across different resolutions for both Imagenet 256 (Figure 7) and Imagenet 512 (above).

## M.2 ADDITIONAL RESULTS ON MULTIMODAL T2I TASKS

**Setup.** To further study the generalizability of the proposed spatial improvements beyond ImageNet, we also perform extensive experiments on multimodal T2I tasks. Following REPA (Yu et al., 2024), we adopt MMDiT (Esser et al., 2024) as the diffusion backbone and apply REPA with various pretrained vision encoders (DINOv2, CLIP, WebSSL, PE, etc.). Same as REPA (Yu et al., 2024), all models are trained for 150K steps with a batch size of 256, and evaluated using an ODE sampler with 50 NFE. Fig. 26 shows the results with REPA before and after application of spatial improvements (iREPA). We observe that the spatial improvements also generalize to multimodal T2I tasks. Furthermore, consistent gains are observed across different choice of pretrained encoders (DINOv2, CLIP, WebSSL, PE, etc.).

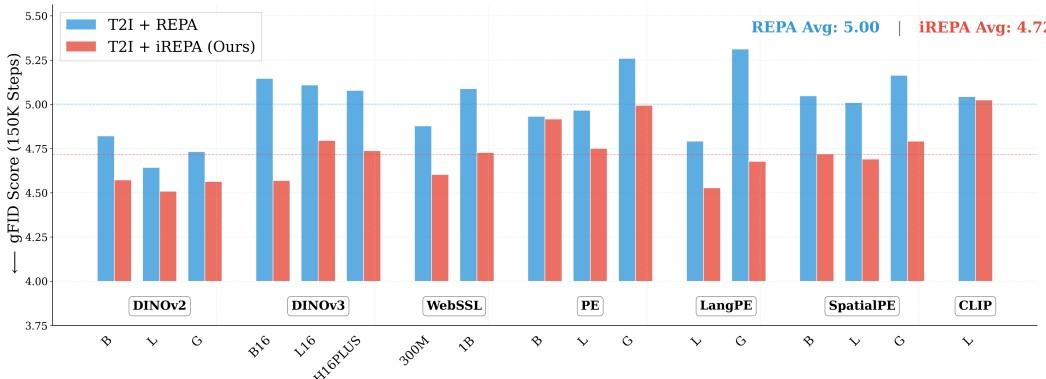

Figure 26: **Text-to-image generation across encoder variants**. Accentuating transfer of spatial features from the target representation to the diffusion features consistently improves convergence speed for both imagenet (refer §4) and multimodal T2I tasks (above). Furthermore, consistent gains are observed across different choice of pretrained encoders (DINOv2, CLIP, WebSSL, PE, etc.).

## M.3    Correlation Analysis Without Outliers (MoCOv3-L and MAE-L)

We repeat the correlation analysis from Section 3 after removing MoCOv3-L and MAE-L, which were identified as outliers. Figure 28 shows the updated correlations across different model sizes. We observe that spatial structure still shows much higher correlation with generation performance over linear probing accuracy. Interestingly, after removing the outliers linear probing actually shows a small positive correlation with gFID (i.e., as linear probing performance increases, the generation becomes worse). This trend is consistent with the observations discussed in Sec. 2, wherein often target representations with higher global semantic performance (linear probing accuracy) show similar or worse generation performance with representation alignment (REPA).

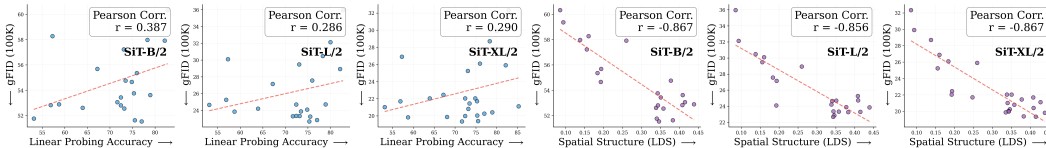

Figure 27: **Correlation analysis without outliers**. Across all model sizes (B, L, XL) spatial structure still shows much higher correlation (Pearson $|r| > 0.85$) with generation performance over linear probing accuracy (Pearson $|r| < 0.38$) after removing the outliers (MoCOv3-L and MAE-L). Interestingly, after removing the outliers linear probing actually shows a small positive correlation with gFID (i.e., as linear probing performance increases, the generation becomes worse). This trend is consistent with the observations discussed in Sec. 2, wherein often target representations with higher global semantic performance (linear probing accuracy) show similar or worse generation performance with representation alignment (REPA).

## M.4 Additional Qualitative Results

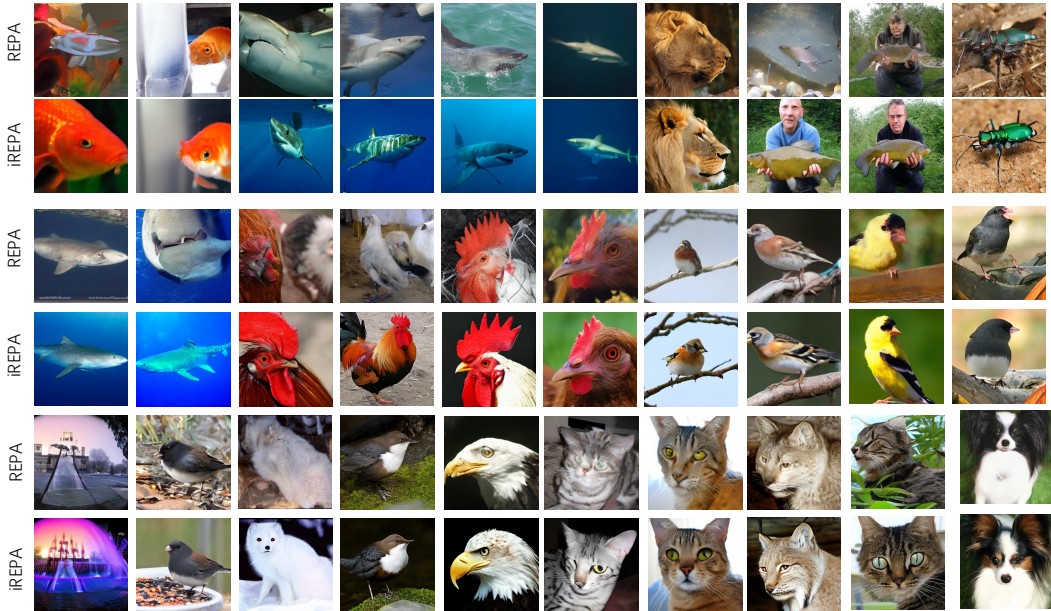

Figure 28: **Additional Qualitative Results** comparing generation outputs before and after application of spatial improvements with REPA. All results are reported using PE-G (Bolya et al., 2025) as the pretrained vision encoder, 400K steps (80 epochs) and with classifier-free guidance scale of 4.0. Similar to quantitative improvements (§4), we also observe that spatial improvements (iREPA) also help improve the visual quality and coherence of the generated outputs.

## M.5 ADDITIONAL RESULTS WITH ALTERNATIVE EVALUATION METRICS

In addition to traditional evaluation metrics (Inception Score, FID, sFID, Precision, Recall), we also verify the robustness of the proposed findings with alternative evaluation metrics for generation quality (Stein et al., 2023; Yang et al., 2023; Kynkäänniemi et al., 2023; Jayasumana et al., 2024). In particular, we use the CMMD metric (Jayasumana et al., 2024) with the PyTorch implementation from (Paul, 2024). Standard reference set from OpenAI ADM evaluation suite (Dhariwal & Nichol, 2021) is used as reference images. We next verify the robustness of the key findings from both §3 and §4 with the CMMD metric.

**Spatial structure correlates much higher with generation performance than linear probing.** To study robustness of analysis from §3, we repeat the large-scale correlation analysis across different vision encoders with CMMD metric (Jayasumana et al., 2024). Results are shown in Figure 29. All results are reported using SiT-B/2 (100K steps) and REPA. All spatial metrics show much higher correlation (Pearson $|r| > 0.88$) with generation performance (CMMD) than linear probing (Pearson $|r| = 0.074$) — demonstrating that key empirical findings from §3 are robust to the choice of evaluation metric (FID or CMMD).

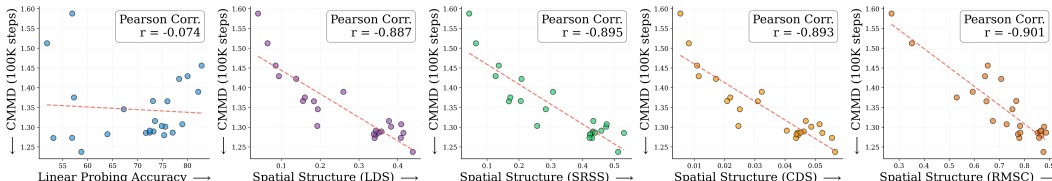

Figure 29: **Correlation analysis with CMMD metric**. We repeat the correlation analysis from §3 with CMMD metric (Jayasumana et al., 2024) instead of gFID. All results are reported using SiT-B/2 (100K steps) and REPA. All spatial metrics show much higher correlation (Pearson $|r| > 0.88$) with generation performance (CMMD) than linear probing (Pearson $|r| = 0.074$). This demonstrates that key empirical findings from §3 are robust to the choice of evaluation metric.

**Accentuating transfer of spatial features helps improve generation quality.** We next study the generalizability of analysis from §4 with the CMMD (Jayasumana et al., 2024) metric in addition to traditional evaluation metrics (gFID, sFID, IS, etc.). Results are shown in Table 8. Across various vision encoders, accentuating transfer of spatial features (iREPA) helps improve convergence speed with both CMMD and traditional evaluation metrics (IS, FID, sFID, Prec., Recall). Furthermore, consistent improvements are observed both with and without classifier-free guidance (CFG).

| Vision Encoder | Steps | w/o CFG | | | | | | w/ CFG | | | | | |
| --- | --- | --- | --- | --- | --- | --- | --- | --- | --- | --- | --- | --- | --- |
| | | CMMD↓ | IS↑ | FID↓ | sFID↓ | Prec.↑ | Rec.↑ | CMMD↓ | IS↑ | FID↓ | sFID↓ | Prec.↑ | Rec.↑ |
| DINOv2-B | 100K | 0.702 | 69.20 | 19.3 | 5.89 | 0.64 | 0.61 | 0.529 | 157.2 | 6.35 | 5.91 | 0.77 | 0.54 |
| +iREPA | 100K | **0.652** | **77.92** | **16.9** | 6.26 | **0.66** | **0.61** | **0.484** | **179.3** | **5.15** | 6.23 | **0.78** | **0.54** |
| DINOv2-B | 400K | 0.455 | 127.4 | 7.76 | 5.06 | 0.70 | 0.66 | 0.320 | 263.0 | 1.98 | 4.60 | 0.80 | 0.61 |
| +iREPA | 400K | **0.438** | **128.6** | **7.52** | **4.89** | **0.71** | **0.65** | **0.310** | **268.8** | **1.93** | **4.59** | **0.80** | 0.60 |
| DINOv3-B | 100K | 0.749 | 63.64 | 21.4 | 6.14 | 0.63 | 0.60 | 0.571 | 144.0 | 7.57 | 6.09 | 0.76 | 0.53 |
| +iREPA | 100K | **0.651** | **78.79** | **16.2** | 6.14 | **0.66** | **0.61** | **0.481** | **181.9** | **4.87** | 6.10 | **0.78** | **0.55** |
| DINOv3-B | 400K | 0.474 | 126.7 | 8.10 | 5.06 | 0.70 | 0.66 | 0.336 | 261.2 | 1.99 | 4.58 | 0.80 | 0.61 |
| +iREPA | 400K | **0.441** | **132.9** | **7.13** | **4.93** | **0.71** | **0.66** | **0.314** | **272.4** | **1.89** | 4.58 | **0.80** | 0.60 |
| WebSSL-1B | 100K | 0.825 | 53.87 | 25.5 | 6.57 | 0.61 | 0.59 | 0.622 | 124.0 | 9.59 | 6.37 | 0.76 | 0.52 |
| +iREPA | 100K | **0.653** | **77.47** | **16.6** | 6.18 | **0.66** | **0.61** | **0.484** | **177.6** | **5.09** | 6.11 | **0.79** | **0.54** |
| WebSSL-1B | 400K | 0.512 | 116.9 | 9.39 | 5.14 | 0.70 | 0.64 | 0.354 | 250.5 | 2.24 | 4.61 | 0.81 | 0.58 |
| +iREPA | 400K | **0.445** | **130.8** | **7.48** | **4.91** | 0.70 | **0.65** | **0.311** | **271.6** | **1.90** | 4.58 | 0.80 | **0.61** |
| PE-Core-G | 100K | 0.922 | 42.74 | 32.3 | 6.70 | 0.57 | 0.59 | 0.714 | 97.2 | 14.1 | 6.56 | 0.71 | 0.53 |
| +iREPA | 100K | **0.697** | **75.01** | **18.1** | 6.03 | **0.64** | **0.61** | **0.525** | **176.8** | **5.66** | 6.08 | **0.77** | **0.54** |
| PE-Core-G | 400K | 0.525 | 109.4 | 10.4 | 5.00 | 0.69 | 0.64 | 0.366 | 238.4 | 2.44 | 4.57 | 0.81 | 0.59 |
| +iREPA | 400K | **0.458** | **132.0** | **7.78** | 5.02 | **0.70** | **0.65** | **0.322** | **275.4** | **1.93** | 4.59 | 0.80 | **0.61** |
| CLIP-L | 100K | 0.790 | 54.07 | 25.2 | 6.65 | 0.61 | 0.60 | 0.605 | 124.5 | 9.77 | 6.59 | 0.75 | 0.52 |
| +iREPA | 100K | **0.657** | **74.46** | **17.8** | 6.33 | **0.65** | **0.61** | **0.489** | **172.0** | **5.67** | 6.33 | **0.78** | **0.54** |
| CLIP-L | 400K | 0.487 | 117.8 | 8.97 | 4.98 | 0.70 | 0.65 | 0.339 | 258.7 | 2.15 | 4.64 | 0.81 | 0.59 |
| +iREPA | 400K | **0.442** | **130.1** | **7.42** | 5.03 | **0.71** | **0.65** | **0.307** | **271.8** | **1.96** | 4.64 | 0.80 | **0.61** |

Table 8: **Additional results with alternative evaluation metrics**. We provide additional results with the CMMD metric (Jayasumana et al., 2024). All results are reported using SiT-XL/2, 256 batch size and traditional REPA as baseline. We adopt the pytorch implementation from (Paul, 2024) for computing the CMMD metric. Standard reference set from OpenAI ADM evaluation suite (Dhariwal & Nichol, 2021) is used as reference images. Across various vision encoders, accentuating transfer of spatial features (iREPA) helps improve convergence speed with both CMMD and traditional evaluation metrics (IS, FID etc.). This demonstrates that key empirical findings from §4 are robust to the choice of evaluation metric.

## M.6 SPATIAL IMPROVEMENTS WITH SAM2

Table 9 and Figure 30 study the impact of spatial normalization (§4) on SAM2-S (46M) encoder features[2]. As shown in Table 9, we observe that for SAM2, while spatial normalization layer helps improve performance, the improvements can be marginal. This is because spatial normalization relies on removing the global component (mean of patch tokens) to enhance spatial contrast. Since SAM2 already has little to no global information (validation accuracy $< 24\%$), spatial normalization only slightly improves the spatial contrast (see Figure 30).

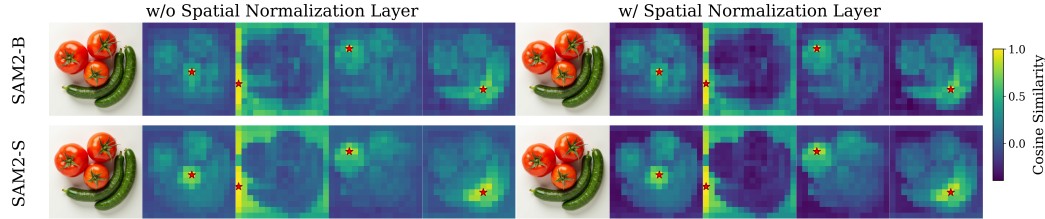

Figure 30: **Impact of spatial normalization on SAM2**. We observe that for SAM2, while use of spatial normalization layer does help enhance the spatial contrast, the improvements can be marginal. This is because spatial normalization (§4) relies on removing the global component (mean of patch tokens) to enhance spatial contrast. Since SAM2 already has little to no global information (validation accuracy $< 24\%$), spatial normalization only slightly improves the spatial contrast.

| Vision Encoder | Steps | IS↑ | FID↓ | sFID↓ | Prec.↑ | Rec.↑ |
|---|---|---|---|---|---|---|
| SAM2-S | 100K | 50.69 | 25.32 | 6.52 | 0.631 | 0.588 |
| **+iREPA** | 100K | **53.14** | **24.52** | **6.28** | **0.629** | **0.591** |
| SAM2-S | 400K | 110.77 | 9.54 | 4.93 | 0.699 | 0.638 |
| **+iREPA** | 400K | **114.62** | **9.10** | **4.89** | **0.704** | **0.640** |

Table 9: **Impact of spatial improvements on SAM2.**. All results are reported with SiT-XL/2, without classifier-free guidance, SAM2-S (46M) as vision encoder and with traditional REPA as the baseline.

---

[2]Note that similar to (Bolya et al., 2025), we use the intermediate output of vision encoder for SAM2 features and not the mask logits. As shown in (Bolya et al., 2025), while mask logits lead to sharper spatial maps, the mask logits itself are not suitable as a target representation.

## M.7 ADDITIONAL RESULTS WITH FULL-FINETUNING ACCURACY INSTEAD OF LINEAR PROBING ACCURACY

**Correlation analysis with full-finetuning accuracy instead of linear probing accuracy**. We repeat the correlation analysis from §3 with the validation accuracy after full-finetuning[3] instead of linear probing. Results are shown in Fig. 31. All results are reported using SiT-XL/2 (100K steps) and REPA. For full-finetuning validation accuracy, instead of linear probing, we perform full-finetuning of vision encoder with learning rate of $5e-5$, warmup ratio of 0.1, and total of 30 epochs. All spatial metrics show much higher correlation with generation performance than full-finetuning accuracy. Interestingly, gFID actually shows a weak positive correlation with the validation accuracy after full-finetuning (Pearson $r = 0.317$), i.e., as validation accuracy increases, the gfid increases, and generation performance becomes worse. This is consistent with the observations discussed in §2, wherein often target representations with higher global semantic performance show similar or worse generation performance with representation alignment (REPA).

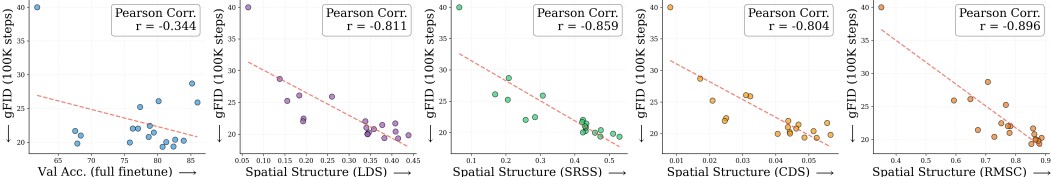

Figure 31: **Correlation analysis with full-finetuning accuracy instead of linear probing accuracy**. We repeat the correlation analysis from §3 with the validation accuracy after full-finetuning instead of linear probing. All results are reported using SiT-XL/2 (100K steps) and REPA. All spatial metrics show much higher correlation with generation performance than full-finetuning accuracy. Interestingly, gFID actually shows a weak positive correlation with the validation accuracy after full-finetuning (Pearson $r = 0.317$), i.e., as validation accuracy increases, the gfid increases, and generation performance becomes worse. This is consistent with the observations discussed in §2, wherein often target representations with higher global semantic performance show similar or worse generation performance with representation alignment (REPA).

---

[3]Please note that while the final findings remain similar, in context of REPA linear probing might be more accurate for estimating global information in *"pretrained"* vision encoders. This is because REPA uses the *"pretrained"* encoder features themselves for regularization. Finetuning the encoder itself can impact the amount of global information. Same as REPA (Yu et al., 2024), we therefore use linear probing accuracy as default for measuring global information.

## M.8    ADDITIONAL RESULTS ON PIXEL-SPACE DIFFUSION MODELS

| Method | #Params | IS↑ | FID↓ | sFID↓ | Prec.↑ | Rec.↑ |
|---|---|---|---|---|---|---|
| PixelFlow (Chen et al., 2025) | 459M | 24.67 | 54.33 | 9.71 | - | 0.58 |
| PixDDT (Wang et al., 2025b) | 434M | 36.24 | 46.37 | 17.14 | - | 0.63 |
| PixNerd (Wang et al., 2025a) | 458M | 43.01 | 37.49 | 10.65 | - | 0.62 |
| DeCo w/o $\mathcal{L}_{\text{FreqFM}}$ | 426M | 46.44 | 34.12 | 10.41 | - | 0.64 |
| DeCo + REPA  (Ma et al., 2025b) | 426M | 48.35 | 31.35 | 9.34 | - | 0.65 |
| JiT (Li & He, 2025a) | 459M | 29.37 | 49.06 | 11.21 | 0.40 | 0.62 |
| **JiT+iREPA (Ours)** | 459M | **50.72** | **29.19** | 9.42 | **0.51** | **0.65** |

Table 10: **Generation performance of pixel-space diffusion models.** Despite its simplicity, when combined with JiT (Li & He, 2025a) iREPA outperforms recently proposed state-of-art pixel-space diffusion methods e.g., DeCo (Ma et al., 2025b). All results are reported with 200K training iterations with a batch size of 256 and evaluated using 50-step Euler sampling without classifier-free guidance. Results for DeCo are obtained directly from (Ma et al., 2025b).

