# OpenReview forum: "What matters for Representation Alignment: Global Information or Spatial Structure?"
_ICLR.cc/2026/Conference — ICLR 2026 Poster_

### Official Review · Reviewer_HkpS · 2025-10-24

**Soundness:** 3
**Presentation:** 3
**Contribution:** 3
**Rating:** 6
**Confidence:** 3

**Summary:**

The paper investigates whether global semantic information or spatial/local information of the external encoder contributes more to the generation performance of REPA (Representation Alignment for diffusion models).
First, the authors show that the amount of global semantic information (quantified by linear probing accuracy) of the external encoder may negatively correlates with REPA generation quality (measured by FID).
Second, they introduce a metric for spatial information of the external representations, defined by how much closer patches exhibit higher feature similarity than distant ones, and show that this spatial inductive bias positively correlates with REPA generation performance.
Finally, the authors propose two simple modifications to improve REPA: (1) replacing the MLP projection layer with a convolutional projection layer to preserve spatial structure, and (2) introducing a spatial normalization to encourage feature consistency among nearby patches. These enhancements consistently improve generation performance across multiple external encoders.

**Strengths:**

- The studied problem is well-motivated:analyzing what are the important factor in REPA of diffusion models.
- The analysis is comprehensive. span across a lot of diverse external recent advanced external encoders and various sizes of diffusion models.
- The empirical findings are clear and well-supported: global semantic information(as measured by linear-probe accuracy) weakly correlates with REPA generation performance(measured by FID), while spatia structure(measured by the spatial metrics) strongly correlates.
-  The simple adjustments proposed in Section 3 are well-motivated, lightweight, and lead to consistent FID improvements across all encoders.
- The qualitative examples in figure 2 gives a nice intuition on the spatial information of generated image.
- The writing is clear and well-structured, making the analysis easy to follow despite the large experimental scope.

Overall, the work provides practical and actionable insights for improving REPA performance in diffusion-based generative models.

**Weaknesses:**

- Figures 2, 3a, and 3c all show a negative trend between global information and generation quality, and Sections 1–2 carry that narrative. But Figure 4 actually shows the correlation is weakly positive among the diverse encoders, as opposed to completely negative. The paper’s wording early on makes it sound stronger than what the data supports.
 - The paper should mention prior findings that REPA tends to improve generation quality at first and then degrade after more training epochs[1] That paper already pointed out that "REPA injects semantic anchors but underuses structural information, which overlaps a lot with this paper’s main claim.
- Defining the linear probing accuracy as "global information" is not entirely intuitive as global information often refer to high-frequency features in respresentation learning domain. "Semantic information" is a better way to describe it.
- Missing details in experimental setup: Figure 3 is lacking some experimental setup details:for example, how many diffusion steps were used, or what diffusion backbone used. These details are important for interpreting the results and comparing across models.

[1]Wang, Ziqiao, et al. "REPA Works Until It Doesn't: Early-Stopped, Holistic Alignment Supercharges Diffusion Training." arXiv preprint arXiv:2505.16792 (2025).

**Questions:**

- Have you tested whether your proposed spatial normalization or convolutional projection still helps when using already spatially rich encoders like SAM2?
- Can you discsuss how doe the previous work's insight situates with your findings? [1]
- Could you clarify the experimental setup for Figure 3: specifically, how many diffusion steps were used and which diffusion backbone the experiments were based on?

[1]Wang, Ziqiao, et al. "REPA Works Until It Doesn't: Early-Stopped, Holistic Alignment Supercharges Diffusion Training." arXiv preprint arXiv:2505.16792 (2025).

---

> ### Author Response · Authors · 2025-11-21
>
> We thank the reviewer for their positive feedback and thoughtful comments. We are excited that the reviewer found our paper having **practical and actionable insights**, **clear and well-motivated**, and with **comprehensive analysis and well-supported findings**.
>
> > Figures 2, 3a, and 3c all show a negative trend between global information and generation quality, and Sections 1–2 carry that narrative. But Figure 4 actually shows the correlation is weakly positive among the diverse encoders, as opposed to completely negative. The paper's wording early on makes it sound stronger than what the data supports.
>
>
> * Yes. On aggregate across all 27 vision encoders, global information does show small negative correlation with generation quality (pearson |r| < 0.26). In comparison, all spatial metrics show much higher correlation with generation quality (pearson |r| > 83). We will revise the wording to reflect this (L132-134).
> * Additionally as pointed by Reviewer S2CT, we observe that after removing the outliers (MoCOv3-L and MAE-L), the correlation between global information and generation quality actually becomes slightly positive (pearson |r| = 0.38). i.e., as linear probing performance increases, gfid also increases and generation becomes worse. This trend is consistent with the observations discussed in Sec. 2., wherein often target representations with higher global semantic performance (linear probing accuracy) show similar or worse generation performance with representation alignment (REPA).
>
> > Can you discuss how doe the previous work's insight situates with your findings? [1]
>
> Interesting question. While we agree with empirical results of [1], we note that [1] hypothesizes that "REPA predominantly distills global semantic information while leaving structural information untapped". The following quote from [1] reflect this hypothesis:
>
> * "*While feature alignment (REPA) accelerates the learning process by injecting global semantic anchors, the structural knowledge remains underexploited.*"
>
> In contrast, surprisingly, we find that spatial structure (not global semantic information) actually already forms one of the key driving factors for the effectiveness of REPA (Sec. 2,3).
>
> Thus while we agree that spatial structure is not fully utilized by REPA (refer Sec. 4), we find that spatial structure (not global semantic information) already plays a very significant role in the effectiveness of REPA. We have revised the paper to include this discussion.
>
> > **Terminology issue:** Defining the linear probing accuracy as "global information" is not entirely intuitive as global information often refer to high-frequency features in respresentation learning domain. "Semantic information" is a better way to describe it.
>
> Thanks, we have revised the manuscript to say "global semantic information" instead of "global information" to make it more clear.
>
>
> > Have you tested whether your proposed spatial normalization or convolutional projection still helps when using already spatially rich encoders like SAM2?
>
> Yes, we find that spatial improvements (iREPA) help improve performance with already spatially rich encoders like SAM2 and SpatialPE (spatially tuned variant of Perceptual Encoders [2]).
>
> * We provide results across all versions of SpatialPE (B,L,G) and various model scales (B,L,XL) in Figure 8, 11, 12 and Table 5. Accentuating transfer of spatial features (iREPA) helps consistently improve convergence speed.
> * We also provide additional results with SAM2-S (46M) in Appendix M.6. Results are shown in Table 9. For SAM2, we observe that while spatial normalization layer helps improve performance, the improvements can be marginal. This is because spatial normalization relies on removing the global component (mean of patch tokens) to enhance spatial contrast. Since SAM2 already has little to no global information (validation accuracy < 24%), spatial normalization only slightly improves the spatial contrast (Figure 29).
>
> > Missing details in experimental setup: Figure 3 is lacking some experimental setup details:for example, how many diffusion steps were used, or what diffusion backbone used. These details are important for interpreting the results and comparing across models.
>
> Thanks. We have revised the caption to also include these details.
> Following REPA, all correlation results are reported using SiT-XL, 250 NFE and without classifier free guidance.
>
> **References:**
>
> [1] Wang, Ziqiao, et al. "REPA Works Until It Doesn't: Early-Stopped, Holistic Alignment Supercharges Diffusion Training." arXiv preprint arXiv:2505.16792 (2025).
>
> [2] Bolya, David, et al. "Perceptual encoders: a new family of vision models." arXiv preprint arXiv:2505.16792 (2025).

---

### Official Review · Reviewer_fK5t · 2025-10-25

**Soundness:** 3
**Presentation:** 3
**Contribution:** 3
**Rating:** 6
**Confidence:** 4

**Summary:**

This work presents an empirical study and enhancement of representation alignment (REPA), which refers to a constraint that aligns a generative model’s representations with those of a pre-trained model during learning. Specifically, the authors extensively experiment with various pre-trained encoders and conclude that representations with “spatially structured” properties contribute more to generation quality than those excelling in classification tasks. The spatial structure is measured using the pairwise cosine similarity between patch tokens. Inspired by this finding, the authors propose a strategy to replace the MLP in REPA with convolutional layers and introduce spatial normalization layers, thereby encouraging the generative model to better learn from the spatial structure of the encoder’s representations.

**Strengths:**

1. The paper systematically examines the relationship between the spatial structure of representations and metrics such as FID by testing an extensive set of encoders, and further strengthens this conclusion using Pearson correlation coefficients.

2. Building on the above findings, the authors propose two strategies to encourage more effective learning of the spatial structure of representations. Both approaches are visually supported through attention map visualizations and are substantiated by ablation studies.

**Weaknesses:**

1. The conclusions of this work rely on the premise that metrics such as FID and IS accurately reflect model generation capability. However, several prior studies[1,2,3, 4] have argued and demonstrated that FID/IS may not fully capture model performance. This introduces some uncertainty to the findings of this work. Thus, it is recommended that the authors provide additional evaluation results based on other metrics - for instance, by replacing the feature extractor used for FID computation.

2. Regarding the experimental setup, this work conducts training and evaluation exclusively on ImageNet at 256×256 resolution. The absence of results on other datasets or at different resolutions (e.g., 512×512) limits the generalizability of the claims. Moreover, since the Inception and VGG models used for FID/sFID/IS/Precision/Recall are also pre-trained on ImageNet, this further compounds the uncertainty in the evaluation.

3. Given the potential limitations of the quantitative evidence raised in the previous points, the paper is notably lacking in qualitative results. Neither generated results of the proposed iREPA method nor qualitative comparisons with other approaches are provided, which would help better assess the actual improvements in generation quality.

4. In the ablation study table (Table 2), under the DINOv2/DINOv3 settings, the sFID results for iREPA (full) are not the best. The authors are expected to explain this observation. Additionally, it is recommended not to format the specific numbers in bold to avoid misunderstanding.

[1] Stein, George, et al. "Exposing flaws of generative model evaluation metrics and their unfair treatment of diffusion models." Neurips 2023.

[2] Yang, Ceyuan, et al. "Revisiting the evaluation of image synthesis with GANs." Neurips 2023.

[3] Kynkäänniemi, Tuomas, et al. "The role of imagenet classes in Fr\'echet Inception Distance." ICLR 2023.

[4] Jayasumana, Sadeep, et al. "Rethinking FID: Towards a better evaluation metric for image generation." CVPR 2024.

**Questions:**

1. My primary concern revolves around the uncertainty introduced by the evaluation metrics, which affects the reliability of the findings and conclusions. The authors may consider addressing this by incorporating the suggestions outlined in the Weaknesses section, such as providing quantitative results with alternative evaluation metrics and on other datasets, as well as including qualitative results to better substantiate the claims.

2. Would the use of encoders trained on segmentation or depth prediction tasks lead to better performance? Intuitively, such representations are likely to exhibit stronger spatial structure, and it would be valuable to examine whether they bring further improvements in the proposed framework.

3. This is an open discussion, not a must: From the perspective of learning spatial structure, would it be beneficial to not apply the REPA constraint at all denoising steps?

---

> ### Author Response · Authors · 2025-11-21
>
> We thank the reviewer for their positive feedback and thoughtful suggestions.
>
> > 1. My primary concern revolves around the uncertainty introduced by the evaluation metrics, which affects the reliability of the findings and conclusions. The authors may consider addressing this by incorporating the suggestions outlined in the Weaknesses section, such as providing quantitative results with alternative evaluation metrics and on other datasets, as well as including qualitative results to better substantiate the claims.
>
>
> Thanks for the suggestion. To study the robustness of the proposed findings, we have revised the manuscript (Appendix M) with several additional experiments and analysis including:
> 1. **Additional quantitative results with alternative metrics** such as CMMD [4] (Appendix M.5)
>
>       a) We repeat the correlation analysis from Sec. 3 with CMMD metric [4] (Figure 26) showing that spatial structure still shows much higher correlation (Pearson $|r| > 0.88$) with generation performance (CMMD) than linear probing (Pearson $|r| = 0.074$).
>
>       b) We also provide additional results with the CMMD metric [4] with iREPA; showing that accentuating transfer of spatial features (iREPA) helps improve convergence speed with both CMMD and traditional evaluation metrics (IS, FID etc.).
> 2. **Additional quantitative results on other datasets** such as MSCOCO (Appendix M.2)
> 3. **Additional quantitative results on other resolutions** (512x512) (Appendix M.1)
> 4. **Additional qualitative results** demonstrating that the proposed spatial improvements also help improve the visual quality and coherence of the generated outputs. (Appendix M.4)
>
> **The results are provided in Appendix M in the revised manuscript.**
>
> We observe that key findings from Sec. 3, 4 are robust to variation in evaluation metrics, datasets, resolutions, and training methods.
>
> > 2. Would the use of encoders trained on segmentation or depth prediction tasks lead to better performance? Intuitively, such representations are likely to exhibit stronger spatial structure, and it would be valuable to examine whether they bring further improvements in the proposed framework.
>
> Yes. Infact, SpatialPE vs PE (Perceptual Encoders) comparison illustrates exactly this point. SpatialPE is a variant derived from PE (Perceptual Encoders) which is tuned for spatial tasks. Results are provided in Figure 2 and Sec. 2. While SpatialPE-B shows a much worse validation accuracy than PE-G (53.1% vs. 82.8%), it leads to better generation with REPA.
>
>
> > 3. This is an open discussion, not a must: From the perspective of learning spatial structure, would it be beneficial to not apply the REPA constraint at all denoising steps?
>
> Excellent question. We believe so. Since spatial structure seems to be more dominant factor, we believe that using a non-uniform timestep sampling for application of REPA loss could help further improve generation quality. We leave further exploration in this direction as future work.
>
> > In the ablation study table (Table 2), under the DINOv2/DINOv3 settings, the sFID results for iREPA (full) are not the best. .. it is recommended not to format the specific numbers in bold to avoid misunderstanding
>
> Thanks. We have updated the paper to remove bold for these two numbers in ablation study in Table 2.
> Also please note that for Dinov3, the sFID for iREPA (6.14) is still better then REPA (6.19); with iREPA (w/o spatialnorm) surprisingly achieving slightly better sFID (5.81).
>
>
>
> **References:**
>
> [1] Stein, George, et al. "Exposing flaws of generative model evaluation metrics and their unfair treatment of diffusion models." Neurips 2023.
>
> [2] Yang, Ceyuan, et al. "Revisiting the evaluation of image synthesis with GANs." Neurips 2023.
>
> [3] Kynkäänniemi, Tuomas, et al. "The role of imagenet classes in Fr'echet Inception Distance." ICLR 2023.
>
> [4] Jayasumana, Sadeep, et al. "Rethinking FID: Towards a better evaluation metric for image generation." CVPR 2024.

---

> > ### Comment · Reviewer_fK5t · 2025-11-26
> > **Reviewer Response to Author Rebuttal**
> >
> > Thank you for the response. I particularly appreciate the additional validation results with other metric, as well as the results on other datasets and resolutions - these make the authors' findings much more robust. Additionally, I highly recommend that the authors include the *full fine-tuning accuracy* (rather than *linear probing*) when evaluating global semantic accuracy, and even consider replacing *linear probing* results with these. This is because, as demonstrated in prior self-supervised learning works such as MAE, *linear probing* often underestimates model performance and does not align with the de facto standard of practical downstream applications.

---

> > > ### Author Response · Authors · 2025-11-28
> > >
> > > >Thank you for the response. **I particularly appreciate the additional validation results** with other metric, as well as the results on other datasets and resolutions - **these make the authors' findings much more robust**.
> > >
> > > We are glad that the reviewer appreciates the additional validation results and also notes that the additional results *"make the authors' findings much more robust"*.
> > >
> > > > Additionally, I highly recommend that the authors include the full fine-tuning accuracy (rather than linear probing) when evaluating global semantic accuracy, and even consider replacing linear probing results with these. This is because, as demonstrated in prior self-supervised learning works such as MAE, linear probing often underestimates model performance and does not align with the de facto standard of practical downstream applications.
> > >
> > > Thanks for the suggestion. We repeat the correlation analysis in Sec. 3, with the validation accuracy after full-finetuning instead of linear probing. **Results are provided in Appendix M.7 of revised manuscript**. While full finetuning does lead to higher validation accuracy, the findings of the correlation analysis in Sec. 3 remain similar. All spatial metrics show much higher correlation with generation performance than full-finetuning accuracy. Please refer Appendix M.7 for further details.
> > >
> > > Please note that while the final findings are similar, in context of REPA linear probing might be more accurate for estimating global information in "pretrained" vision encoders. This is because REPA uses the "pretrained" encoder features themselves for regularization. Finetuning the encoder itself can impact the amount of global information. Same as REPA (Yu et al., 2024), we therefore use linear probing accuracy as default for measuring global information. We have added both results with full-finetuning accuracy and this discussion in the revised manuscript.
> > >
> > > We sincerely appreciate the reviewer's suggestions, which have helped strengthen the key findings of the paper. Thank you again for your time and valuable input.

---

> ### Comment · Reviewer_fK5t · 2025-11-28
> **Reviewer Response to Author Rebuttal (2)**
>
> While I have some reservations regarding the experiments on linear probing and full fine-tuning (I now believe it would be better to add the full fine-tuning accuracy rather than replace the existing results), I truly appreciate the authors' willingness and efforts in strengthening the robustness of their research. I intend to raise my rating, yet the editing function is currently restricted. Let’s see whether it will become available again later.

---

### Official Review · Reviewer_S2CT · 2025-10-31

**Soundness:** 3
**Presentation:** 4
**Contribution:** 4
**Rating:** 8
**Confidence:** 3

**Summary:**

First, this paper investigates the underlying factors that make REPA effective. Through a large-scale empirical study involving 27 different vision encoders, the paper demonstrates that spatial structure is more important than global information. Second, the paper introduces a simple and efficient metric, termed Spatial Structure Metric (SSM). Third, the paper proposes an improved iREPA method which is designed to better preserve and transfer spatial information. Extensive experiments show that iREPA consistently accelerates convergence and improves final generation quality over the baseline REPA across a wide variety of encoders, diffusion model sizes, and training variants.

**Strengths:**

S1. The paper's primary contribution—that spatial structure, not global semantic accuracy, is the key driver for REPA's success—is a significant and non-obvious finding. The authors have conducted a comprehensive set of experiments to validate their claims.
S2. The proposed iREPA method is elegantly simple (noted as <4 lines of code) yet highly effective. The two modifications (convolutional projection and spatial normalization) are well-motivated by the paper's core finding and are easy to implement, which significantly increases the practical value and potential for adoption.
S3. The experimental results clearly demonstrate the effectiveness of iREPA across multiple evaluation aspects. Specifically, iREPA exhibits consistent performance gains in scalability (handling larger model sizes), robustness across different encoder depths, and improvements in generation quality.

**Weaknesses:**

W1. The paper primarily focuses on improving REPA and its variants. While this is a valid contribution, the proposed method is not benchmarked against other, orthogonal techniques for improving generative model training.
W2. In the correlation plot between linear probing accuracy and gFID (Fig 1 left), there are two noticeable outliers (points 25: Mocov3-l, 27: MAE-l) that have both low accuracy and poor (high) gFID. These high-leverage points significantly influence the linear regression.

**Questions:**

Q1. Following on from W1, could the authors show how iREPA's convergence acceleration compares to other popular methods for improving generative model training? Could it work together with other training methods?
Q2. If the outliers are removed, will the conclusion corresponding to Figure 1 still hold?
Q3. The iREPA modifications are praised for their simplicity. However, given the strong conclusion that spatial structure is key, did the authors experiment with more sophisticated mechanisms or network structure to maximize spatial information transfer?

---

> ### Author Response · Authors · 2025-11-21
>
> We thank the reviewer for their positive feedback and thoughtful comments. We are excited that the reviewer found our paper to have a **significant contribution and non-obvious finding**. We are also happy that they found our work **elegantly simple yet highly effective**, with **significant practical value** and backed by **comprehensive experiments to support central findings**.
>
> > Q1. Following on from W1, could the authors show how iREPA's convergence acceleration compares to other popular methods for improving generative model training? Could it work together with other training methods?
>
> Yes, while we primarily use iREPA to highlight role of spatial structure for representation alignment, the proposed spatial improvements are complementary to and can be used with other training methods.
>
> Results for generalization to other training recipes are provided in Sec. 4. (L317-319, L412-415) and Tab. 3. We find that accentuating spatial feature transfer helps consistently improve convergence speed across different training recipes such as REPA, REPA-E and MeanFlow.
>
> > Q2. In the correlation plot between linear probing accuracy and gFID (Fig 1 left), there are two noticeable outliers (points 25: Mocov3-l, 27: MAE-l) that have both low accuracy and poor (high) gFID.  ... If the outliers (MoCOv3-L and MAE-L) are removed, will the conclusion corresponding to Figure 1 still hold?
>
> Yes the conclusions still hold. We repeat the correlation analysis from Section 3 after removing MoCOv3-L and MAE-L, which were identified as outliers.
>
> **Results are provided in Appendix M.3 (revised manuscript).**
>
> * We observe that spatial structure still shows much higher correlation (Pearson $|r| > 0.85$) with generation performance over linear probing accuracy (Pearson $|r| < 0.38$) after removing the outliers (MoCOv3-L and MAE-L).
> * Interestingly, after removing the outliers linear probing actually shows a small positive correlation with gFID (i.e., as linear probing performance increases, the generation becomes worse). This trend is consistent with the observations discussed in Sec. 3, wherein target representations with higher global performance (linear probing accuracy) show similar or worse generation performance with representation alignment (REPA).
>
> > Q3. The iREPA modifications are praised for their simplicity. However, given the strong conclusion that spatial structure is key, did the authors experiment with more sophisticated mechanisms or network structure to maximize spatial information transfer?
>
> Thanks for praising the simplicity of the spatial modifications.
> Our paper focuses on a minimalist approach, and we intentionally use very simple modifications to highlight the impact of spatial feature transfer on diffusion model training with minimal changes (<4 lines of code).
>
> In our experiments we also tried the following modifications which provide additional gains.
> 1) Gram matrix loss [1,2] for explicitly encouraging the spatial self-similarity of diffusion features to match the spatial self-similarity of the pretrained vision encoder features (DINOv2, CLIP, etc.).
> 2) Replacing the REPA cosine similarity loss with contrastive loss [3] which encourages more spatial contrast during spatial feature transfer.
>
>
> Please note that the above modifications are still relatively simple and can be further improved with more sophisticated mechanisms. Also note that we do not use this or any additional loss in the experiments presented in this paper.
>
> **References**:
>
> [1] DINOv3, Siméonin et al., 2025.
>
> [2] The Platonic Representation Hypothesis, Huh et al., 2024.
>
> [3] A Simple Framework for Contrastive Learning of Visual Representations, Chen et al., 2020.

---

### Official Review · Reviewer_6Vka · 2025-11-01

**Soundness:** 4
**Presentation:** 4
**Contribution:** 3
**Rating:** 8
**Confidence:** 4

**Summary:**

This paper investigates REPA and systematically analyses how to best guide diffusion model training. With external representations. Through a large-scale empirical study, the authors show that spatial structure correlates more strongly with generation quality than global semantic information and based on these findings, the paper introduces iREPA. The method is a minimal but effective change to the existing pipeline that shows consistent improvements to REPA by replacing the MLP with a convolutional layer and adding spatiall normalization layer to enhance spatial contrast.

**Strengths:**

- **Fundamental insight that provides stable gains**: the work provides large-scale evidence that spatial structure rather than semantic quality determines usefulness of pertained visual features for generative alignment and uses this insight to implement a simple and minimal intervention, that improves convergence and generative quality consistently.
- **Generalization and robustness:** the proposed method is able to show improvements across multiple architectures / encoders (DINOV2, SAM2, CLIP, etc.), model scales (-B to -XL) and training variants
- **Extensive experimentation and validation:** provides extensive correlation studies, ablations, and analyses to support the central claim

**Weaknesses:**

**Effect of removing global semantics:** iREPA intentionally suppresses the global semantic component of pretrained representations to enhance spatial contrast. While this clearly benefits diffusion-based generation, it remains uncertain how much this trade-off might affect tasks that depend on higher-level semantic coherence or multimodal conditioning.

**Questions:**

1. **Generalization to multimodal or semantically guided setups**: Given that iREPA explicitly downweights global semantic signals, how might the approach generalize to multimodal or text-conditioned diffusion models where semantic correspondence is critical? Would some hybrid form of spatial and global alignment be beneficial in that setting?
2. While the results clearly show that preserving spatial correlations improves sample realism, the mechanism behind this effect is not fully explained. Do the authors believe that maintaining local spatial dependencies helps the diffusion model reconstruct structure more coherently during denoising, or that it serves as a regularizing inductive bias for patch-level consistency?

---

> ### Author Response · Authors · 2025-11-21
>
> We thank the reviewer for their positive feedback and thoughtful comments. We are excited that the reviewer found our paper to have **fundamental insights that provide stable gains**, **generalizable across training settings** (encoders,models scales etc) and **backed by extensive experiments to support central findings**.
>
> > Given that iREPA explicitly downweights global semantic signals, how might the approach generalize to multimodal text-conditioned diffusion models where semantic correspondence is critical?
>
> To further study the generalizability of the proposed spatial improvements beyond ImageNet, we also perform extensive experiments on T2I training. Following REPA (Yu et al., 2024), we adopt MMDiT as the diffusion backbone and apply REPA loss across various pretrained vision encoders (e.g., DINOv2, CLIP, WebSSL, PE, etc.). Same as REPA (Yu et al., 2024), all experiments use 150K steps and 256 batch size for training.
>
> **Results for T2I training across various encoders are provided in Appendix M.2 in the revised manuscript.**
>
> Figure 25 (revised manuscript) shows the results with REPA before and after application of spatial improvements (iREPA). We observe that the spatial improvements also generalize to multimodal T2I tasks. Furthermore, consistent gains are observed across different choice of pretrained encoders (DINOv2/3, CLIP, WebSSL, PE, etc.).
>
> > While the results clearly show that preserving spatial correlations improves sample realism, the mechanism behind this effect is not fully explained. Do the authors believe that maintaining local spatial dependencies helps the diffusion model reconstruct structure more coherently during denoising, or that it serves as a regularizing inductive bias for patch-level consistency?
>
> Excellent question. We believe it is closely related to the first hypothesis you mentioned.
>
> Spatial structure provides much more spatially-finegrained inductive bias as compared to high-level global semantic information. We believe that the presence of this finegrained inductive bias is important for generating more structurally coherent images during the diffusion denoising process. We will add this to the revised manuscript.

---

### Author Response · Authors · 2025-11-21

We thank all reviewers for their positive feedback. We are excited that all reviewers found our paper to have **fundamental insights with significant and non-obvious findings** [6Vka, S2CT, HKPS], **backed by extensive experiments to support central claims** [6Vka, S2CT, fK5t, HKPS], and **elegantly simple yet highly effective** [6Vka, S2CT, fK5t, HKPS] across multiple vision encoders (DINOV2/3, WebSSL, PE, CLIP, etc.), model scales (B to XL) and training recipes (REPA, REPA-E, MeanFlow).

We also thank the reviewers for thoughtful suggestions to further strengthen the central findings. We have accordingly revised the manuscript (Appendix M) with additional experiments and analyses including:
* Additional experiments on multimodal T2I tasks across various encoders (DINOv2, DINOv3, CLIP, WebSSL, PE, etc.).
* Additional results at higher resolutions (512x512) across different model scales and vision encoders.
* Analysis showing robustness of central findings to alternative evaluation metrics for generation quality.
* Additional qualitative results before and after application of spatial improvements (iREPA).

---

### Author Response · Authors · 2025-12-03
**Summary of Discussion Phase (P1)**

We thank all reviewers and the Area Chair for their positive feedback. Since the discussion phase is coming to an end, and in light of recent technical issues with the conference platform, we want to provide a brief summary of the discussion phase.

**Initial Scores:**
The paper received strong positive inital scores of 8,8,6,6 before the discussion phase.

We are excited that all reviewers found our paper to have **fundamental insights with significant and non-obvious findings** [6Vka, S2CT, HKPS], **backed by extensive experiments to support central claims** [6Vka, S2CT, fK5t, HKPS], and **elegantly simple yet highly effective** [6Vka, S2CT, fK5t, HKPS] across multiple vision encoders (DINOV2/3, WebSSL, PE, CLIP, etc.), model scales (B to XL) and training recipes (REPA, REPA-E, MeanFlow).

**Summary of Discussion Phase:**

1. **Reviewer 6Vka**
   * **Initial Rating**: 8
   * **Key Strengths**: The reviewer praised that the paper has fundamental insights that provide stable gains, generalizable across training settings (encoders,models scales etc) and backed by extensive experiments to support central findings.
   * **Main Questions and Author Response**:
     - Q1: Does iREPA generalize to multimodal text-conditioned diffusion models?
        - A1: We have revised the manuscript to add extensive T2I experiments on MSCOCO (Appendix M.2, Fig. 25) across DINOv2, CLIP, WebSSL, PE showing consistent gains.
     - Q2: Mechanism behind why spatial matters more then global for representation alignment?
        - A2: We have provided a detailed explanation for why spatial matters more then global for representation alignment.
   * Discussion Outcome: Reviewer could not respond yet due to platform technical issues.



2. **Reviewer S2CT**
   * **Initial Rating**: 8
   * **Key Strengths**: The reviewer praised that the paper provides a significant and non-obvious finding; is elegantly simple yet highly effective; and demonstrates clear empirical findings across multiple evaluations.
   * **Main Questions and Author Response**:
     - Q1: Does iREPA work with other training methods?
         - A1: Yes. We provide results showing generation of spatial improvements to other training recipes such as REPA-E and MeanFlow in Table 3.
     - Q2: Does correlation results hold if outliers (MoCOv3-L, MAE-L) removed?
        - A2: Yes they hold. We repeat the correlation analysis without outliers (Appendix M.3), confirming spatial structure still shows much higher correlation (|r| > 0.85) than linear probing (|r| < 0.38).
     - Q3: iREPA is praised for simplicity .. but have authors tried more sophisticated mechanisms to maximize spatial information transfer?
        - A3: Yes. We discuss additional spatial mechanisms such as gram matrix and contrastive loss in the author response.
   * **Discussion Outcome**: Reviewer could not respond yet due to platform technical issues.

---

> ### Author Response · Authors · 2025-12-03
> **Summary of Discussion Phase (P2)**
>
> 3. **Reviewer fK5t**
>    * **Initial Rating**: 6 (Accept)
>    * **Main Questions and Author Response**:
>      - Q1: The main concern of the reviewer was around robustness of the findings to more diverse setups such as different evaluation metrics, datasets, resolutions, and training methods.
>         - A1: To study the robustness of the findings, we have added additional experiments and analyses for all requested setups including:
>             - a) Additional quantitative results with alternative metrics such as CMMD [4] (Appendix M.5)
>             - b) Additional quantitative results on other datasets such as MSCOCO (Appendix M.2)
>             - c) Additional quantitative results on other resolutions (512x512) (Appendix M.1)
>             - d) Additional qualitative results (Appendix M.4)
>
>        **We find that the findings are robust to the choice of evaluation metric, datasets, resolutions, and training methods.**
>      - Q2: Would use of encoders trained for segmentation or spatial tasks help?
>        - A2: Yes. We provide comparison with SpatialPE vs PE (Perceptual Encoders) in Figure 2 and Sec. 2. demonstrating that encoders tuned of spatial tasks improve generation quality.
>      - Q3: Additionally, I highly recommend that the authors include the full fine-tuning accuracy (rather than linear probing)
>        - A3: We repeat the correlation analysis with the validation accuracy after full-finetuning instead of linear probing. Results are provided in Appendix M.7; showing robustness of the findings.
>    * **Discussion Outcome**: The reviewer highly praised the additional validation results and intended to further raise score mentioning **"I particularly appreciate the additional validation results"** and **"I truly appreciate the authors' willingness and efforts in strengthening the robustness of their research. I intend to raise my rating, yet the editing function is currently restricted."**
>
> 4. **Reviewer HkpS**
>    * **Initial Rating**: 6 (Accept)
>    * **Key Strengths**: The reviewer praised that the paper provides practical and actionable insights, is clear and well-motivated, and has very comprehensive analysis and well-supported findings.
>    * **Main Questions and Author Response**:
>      - Q1: framing of correlation (negative vs weakly positive)
>         - A1: We have revised the manuscript to clarify the correlation framing (weakly positive vs negative).
>      - Q2: How does the findings of this paper situate with the previous work [1]?
>         - A2: [1] hypothesizes that "REPA predominantly distills global semantic information while leaving strcutural information untapped". In contrast, surprisingly, we show that spatial structure (not global semantic information) actually already forms one of the key driving factors for the effectiveness of REPA (Sec. 2,3).
>      - Q3: Terminology issue: "semantic information" is a better way to describe it.
>         - A3: We have revised the manuscript to say "global semantic information" instead of "global information" to make it more clear.
>      - Q4: Testing on spatially rich encoders?
>         - A4: We have added experiments with SAM2-S and SpatialPE, showing that spatial improvements help even with already spatially rich encoders. (Figure 8, 11, 12 and Table 5, 9)
>    * **Discussion Outcome**: Reviewer could not respond yet due to platform technical issues.

---

### Meta-Review · Area_Chair_c1U9 · 2026-01-05

**Summary:**

- Unclear trade-off and mechanism behind suppressing global information. 6Vka mentioned that the mechanism that preserving spatial correlations improves sample realism is not fully explained. HkpS notes that early sections claim a strong negative correlation between global information and generation quality, but “Figure 4 actually shows the correlation is weakly positive among the diverse encoders.”

- Potentially flawed or biased evaluation metrics. fK5t pointed out several prior studies have argued and demonstrated that FID/IS may not fully capture model performance. Moreover, the paper is notably lacking in qualitative results.

- Limited empirical comparisons. S2CT pointed out the proposed method is not benchmarked against other, orthogonal techniques for improving generative model training. They also noticed that noticeable outliers significantly influence the linear regression.

**Reviewer Concerns:**

The rebuttal provides helpful clarifications and discussion, resolving several core experimental and evaluation concerns raised by multiple reviewers.

**Reviewer Scores:**

6Vka: no score change.

S2CT: no score change.

fK5t: raise score.

HkpS: no score change.

---

### Decision · Program_Chairs · 2026-01-26

Accept (Poster)